# Scalable Best-of-N Selection for Large Language Models via Self-Certainty

**Zhewei Kang**[*]
UC Berkeley
waynekang@berkeley.edu

**Xuandong Zhao**[*†]
UC Berkeley
xuandongzhao@berkeley.edu

**Dawn Song**
UC Berkeley
dawnsong@berkeley.edu

## Abstract

Best-of-N selection is a key technique for improving the reasoning performance of Large Language Models (LLMs) through increased test-time computation. Current state-of-the-art methods often employ computationally intensive reward models for response evaluation and selection. Reward-free alternatives, like self-consistency and universal self-consistency, are limited in their ability to handle open-ended generation tasks or scale effectively. To address these limitations, we propose *self-certainty*, a novel and efficient metric that leverages the inherent probability distribution of LLM outputs to estimate response quality without requiring external reward models. We hypothesize that higher distributional self-certainty, aggregated across multiple samples, correlates with improved response accuracy, as it reflects greater confidence in the generated output. Through extensive experiments on various reasoning tasks, we demonstrate that self-certainty (1) scales effectively with increasing sample size $N$, akin to reward models but without the computational overhead; (2) complements chain-of-thought, improving reasoning performance beyond greedy decoding; and (3) generalizes to open-ended tasks where traditional self-consistency methods fall short. Our findings establish self-certainty as a practical and efficient way for improving LLM reasoning capabilities. The code is available at https://github.com/backprop07/Self-Certainty

## 1 Introduction

Large Language Models (LLMs) have achieved impressive reasoning abilities, yet reliably producing accurate outputs for complex tasks often requires techniques to enhance inference-time performance [Wu et al., 2024, Xiang et al., 2025]. *Best-of-N* selection, generating and selecting from multiple candidate responses, has emerged as a powerful paradigm for significantly improving reasoning accuracy [Snell et al., 2024]. Current Best-of-N methods frequently rely on reward models, such as Outcome Reward Models (ORMs) [Cobbe et al., 2021a] and Process Reward Models (PRMs) [Lightman et al., 2023, Uesato et al., 2022], not only for output selection but also for data annotation to further refine LLM reasoning capabilities [Uesato et al., 2022, Wang et al., 2022].

However, reward models introduce substantial computational and practical challenges. They are computationally expensive to train or fine-tune, often requiring as many parameters as the LLM itself [Wang et al., 2024a], are vulnerable to distribution shifts, and can suffer from "reward hacking" [Eisenstein et al., 2023]. While techniques like reward model ensembles [Coste et al., 2023] offer partial mitigation, they further increase overhead.

As a lighter-weight alternative, self-consistency [Wang et al., 2022] aggregates multiple outputs using majority voting. However, it is applicable only to tasks with directly comparable string-matched

---

[*]Equal contribution
[†]Corresponding author

39th Conference on Neural Information Processing Systems (NeurIPS 2025).

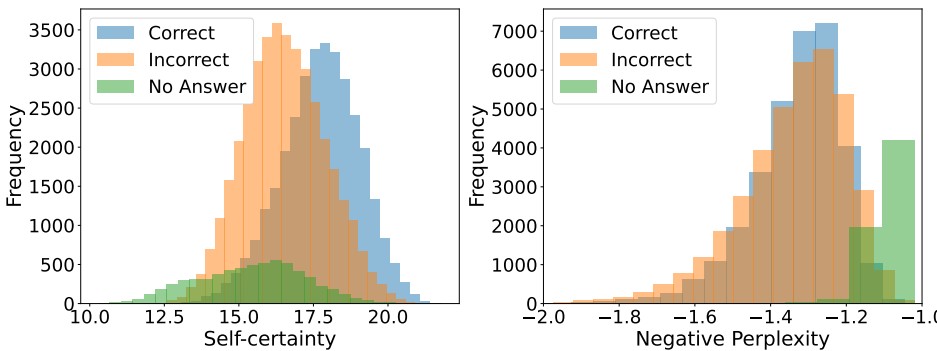

Figure 1: Distribution of self-certainty and negative perplexity for correct, incorrect, and no-answer responses on the MATH dataset (Level 4) [Hendrycks et al., 2021] using the Llama-3.1-8B-Instruct with 64 samples per question. For self-certainty, the distributions of correct and incorrect groups concentrate around different central values, with frequencies decreasing smoothly toward both extremes. In contrast, negative perplexity fails to clearly separate correct from incorrect outputs and favors no-answer responses, highlighting self-certainty's effectiveness in assessing response quality.

answers, limiting its use for differentiating reasoning paths or open-ended tasks. Universal self-consistency (USC) [Chen et al., 2023] prompts the LLM to choose the most consistent response, but its gains are constrained by context length and model ability, sometimes declining with larger $N$ [Cobbe et al., 2021b], and can be ineffective for small models, as our research confirms. Moreover, self-consistency and USC lack a direct quality score for responses, limiting their applicability in tasks such as candidate ranking.

To overcome these limitations, we propose leveraging the LLM's inherent probabilistic output for a more practical, general, and robust approach to Best-of-N selection. We hypothesize that an LLM's probability distribution naturally encodes its *certainty*. We introduce **self-certainty**, a novel metric quantifying this confidence by measuring the divergence of the predicted token distribution from a uniform distribution. A distribution diverging significantly from uniform indicates a more peaked—and thus more certain—prediction. As shown in Figure 1, self-certainty demonstrates a stronger signal for distinguishing correct responses. Notably, it incurs almost no computational overhead, as the token distribution is generated alongside the tokens during inference. Inspired by Borda Voting, we enhance self-consistency by incorporating self-certainty-based ranking, assigning weighted votes based on self-certainty rank using a scaling factor of $(N - \text{ranking} + 1)^p$, effectively prioritizing more confident responses.

We rigorously evaluate our methods across diverse reasoning benchmarks, including LiveBench-Math [White et al., 2024], GSM8K [Cobbe et al., 2021b], MATH [Hendrycks et al., 2021], CRUXEval [Gu et al., 2024] and LiveCodeBench [Jain et al., 2024], spanning mathematical reasoning, code reasoning, and code generation. Our experiments reveal that self-certainty-based voting consistently outperforms self-consistency in Best-of-N selection of reasoning tasks, effectively adapting to varying sample sizes and question difficulties.

The key advantages of self-certainty are:

- **Scalability**: Self-certainty scales efficiently with increasing sample size $N$, mirroring reward models in scalability but without their computational burden.
- **Orthogonal Enhancement to Chain-of-Thought**: Self-certainty complements chain-of-thought (CoT) reasoning [Wei et al., 2022], outperforming self-consistency through weighted voting.
- **Generalizability to Open-Ended Tasks**: Self-certainty generalizes effectively to open-ended responses (e.g., code) where self-consistency is inapplicable, surpassing greedy decoding and USC.

## 2   Related Works

**Reward Models for Response Reranking and Selection.**   Evaluating LLM outputs with external models like verifiers or reward models (ORMs, PRMs) can enhance reasoning and select best

samples [Lightman et al., 2023, Wang et al., 2024a]. However, these models are often task-specific, sensitive to the base model [Eisenstein et al., 2023], and computationally expensive to train, sometimes requiring parameter counts similar to the LLMs they evaluate [Wang et al., 2024a]. Our approach, self-certainty, avoids additional training by using the LLM's own logits for efficient quality assessment.

**Consistency-Based Response Selection.** Self-consistency [Wang et al., 2022] leverages the model's internal understanding by selecting the most common response from multiple outputs, improving reliability. However, it's limited to tasks with convergent final answers and hard to generalize to open-ended generation. universal self-consistency (USC) [Chen et al., 2023] extends to more tasks but faces scalability issues and lacks a certainty measure. Self-certainty overcomes these limitations by directly measuring response confidence from token distributions, handling open-ended tasks and scaling efficiently.

**Confidence Estimation for Model Responses.** Various methods estimate model confidence [Geng et al., 2023]. Self-Evaluation [Ren et al., 2023] uses yes/no token probabilities. BSDetector [Chen and Mueller, 2024] measures similarity and prompts for self-verification. TrustScore [Zheng et al., 2024] computes likelihood against modified-prompt distractors. These often require multiple evaluations, hindering scalability for Best-of-N selection. In contrast, self-certainty leverages the output token distribution directly, avoiding extra prompts and enabling efficient, scalable selection.

# 3 Measuring Confidence of LLMs

This section explores metrics for quantifying LLM prediction confidence, comparing probabilistic measures with distributional ones to identify the most effective for reliable output selection.

## 3.1 LLM Background

Large Language Models, typically Transformer-based [Vaswani, 2017], autoregressively generate token sequences $y = (y_1, \ldots, y_m)$ from an input $x = (x_1, \ldots, x_n)$. At each step $i$, the model produces logits $\ell_i \in \mathbb{R}^V$ (where $V = |\mathcal{V}|$ is vocabulary size), which convert to a probability distribution $p(\cdot|x, y_{<i}) \in [0, 1]^V$ over the vocabulary for the next token $y_i$. This distribution reflects the model's belief about the next token.

## 3.2 Sentence-Level Probabilistic Confidence

Probabilistic confidence quantifies a model's certainty in its predictions by directly leveraging the probabilities assigned to sampled tokens.

**Average Log-Probability.** A common confidence measure is the average log-probability (AvgLogP) of sampled tokens:

$$\text{AvgLogP} \coloneqq \frac{1}{n} \sum_{i=1}^{n} \log \left[ p(y_i|x, y_{<i}) \right] \tag{1}$$

where $p(y_i|x, y_{<i})$ is the probability of token $y_i$. Higher AvgLogP values indicate the model assigns higher probabilities to generated tokens, reflecting greater confidence.

**Perplexity.** Perplexity is a common metric for evaluating language models, defined as the exponentiated average negative log-likelihood:

$$\text{Perplexity} \coloneqq \exp \left( -\frac{1}{n} \sum_{i=1}^{n} \log \left[ p(y_i|x, y_{<i}) \right] \right) \tag{2}$$

Since Perplexity $= \exp(-\text{AvgLogP})$, both measures are equivalent when selecting responses. We use negative perplexity for Best-of-N selection, though studies show it struggles with long contexts [Hu et al., 2024], suggesting the need for alternatives.

## 3.3 Distributional Confidence

Distributional confidence measures consider the entire probability distribution over the vocabulary at each generation step, capturing a more holistic view of the model's certainty beyond just sampled token probabilities.

A sentence-level distributional confidence measure can be defined as:

$$\text{Distributional-Confidence} := F(f(P_{y|x})) \tag{3}$$

where $P_{y|x} = (p(\cdot|x), p(\cdot|x, y_1), \ldots, p(\cdot|x, y_{<n}))$ represents the sequence of token-level probability distributions, $f$ produces a confidence score for each token, and $F$ aggregates these into a sentence-level confidence. With output length $n$, we define $F$ as the average across all positions:

$$F(C_1, \ldots, C_n) = \frac{1}{n} \sum_{i=1}^{n} C_i, \quad C_i = f(p(\cdot|x, y_{\leq i})) \tag{4}$$

For function $f$, we explore metrics that quantify how "peaked" or "concentrated" the probability distribution is, with more concentrated distributions suggesting higher model certainty:

**Kullback-Leibler (KL) Divergence.** Drawing upon neural networks as Maximum Likelihood Estimators [LeCun et al., 2015], we hypothesize that higher confidence corresponds to distributions further from a uniform distribution $U$ (representing maximum uncertainty). KL Divergence quantifies this difference:

$$C_i^{\text{KL}} := \text{KL}(U \parallel p(\cdot|x, y_{\leq i})) = \sum_{j=1}^{V} \frac{1}{V} \log\left(\frac{1/V}{p(j|x, y_{\leq i})}\right) = -\frac{1}{V} \sum_{j=1}^{V} \log\left(V \cdot p(j|x, y_{\leq i})\right) \tag{5}$$

**Gini Impurity.** Originally introduced in decision trees [Breiman, 2017], Gini Impurity measures the probability that two randomly sampled tokens belong to different classes. A more concentrated distribution indicates higher confidence:

$$C_i^{\text{Gini}} := 1 - I_G(p(\cdot|x, y_{\leq i})) = \sum_{j=1}^{V} (p(j|x, y_{\leq i}))^2 \tag{6}$$

**Entropy.** Entropy measures the disorder in a probability distribution. Higher entropy indicates greater uncertainty, so we use negative entropy as a confidence measure:

$$C_i^{\text{Entropy}} := \sum_{j=1}^{V} p(j|x, y_{\leq i}) \log(p(j|x, y_{\leq i})) \tag{7}$$

**Distributional Perplexity (DP).** We apply a negative sign to perplexity to interpret it as confidence. To distinguish from standard perplexity (Equation 2), we denote it as DP:

$$C_i^{\text{DP}} := -\exp\left(-\sum_{j=1}^{V} p(j|x, y_{\leq i}) \log(p(j|x, y_{\leq i}))\right) \tag{8}$$

## 3.4 Our Primary Metric: Self-Certainty

Empirical evaluations (Figure 1, 4) demonstrate that KL-divergence-inspired distributional confidence more effectively distinguishes correct samples from incorrect ones and achieves superior accuracy at higher $N$ values. Theoretically, an infinitesimal gradient step that increases the log-likelihood of a desired token raises its token-wise self-certainty whenever that token already carries sufficiently high probability. A detailed proof is provided in Appendix A.1. Consequently, standard training procedures, such as supervised fine-tuning and reinforcement learning, that increase the probability of the desired token also tend to increase its self-certainty, supporting our choice of self-certainty as a principled confidence metrics. Based on these findings, we define self-certainty as our primary confidence metric for best-of-$N$ selection:

$$\textbf{Self-Certainty} = -\frac{1}{nV} \sum_{i=1}^{n} \sum_{j=1}^{V} \log\left(V \cdot p(j|x, y_{\leq i})\right) \tag{9}$$

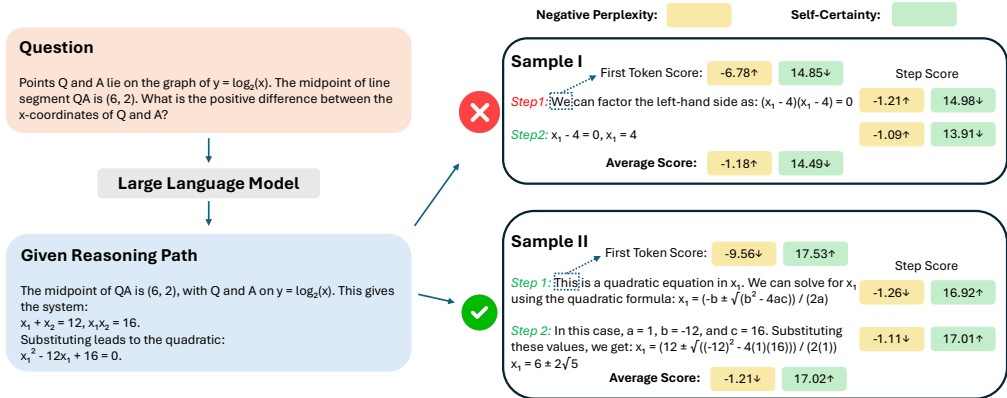

Figure 2: Comparison of reasoning paths in solving a quadratic equation for the given problem using self-certainty and negative perplexity. Sample I factors the quadratic equation directly, while Sample II applies the quadratic formula. The figure illustrates an example of how the two measures assign confidence scores at each reasoning step, showing that self-certainty distinguishes between correct and incorrect reasoning more effectively than negative perplexity.

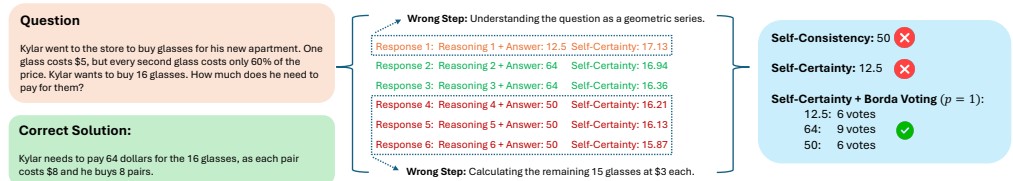

Figure 3: Example of Borda Voting correctly identifying the answer when confidence-driven selection and self-consistency fail. The figure illustrates how Borda Voting aggregates confidence scores and ranks to select the correct answer.

Cross entropy between the predicted distribution and a uniform distribution provides an equivalent confidence measure, differing from KL-divergence only by a constant. The self-certainty based on cross-entropy is:

$$\text{Self-Certainty (CE)} = -\frac{1}{nV}\sum_{i=1}^{n}\sum_{j=1}^{V}\log\bigl(p(j\mid x, y_{\leq i})\bigr). \qquad (10)$$

## 3.5 Analysis

Reward Models (PRMs and ORMs) typically evaluate responses using the minimum reward across all reasoning steps [Lightman et al., 2023, Wang et al., 2024a], prioritizing error detection over progress assessment. Self-certainty methods effectively identify mistakes through averaging because early errors propagate, reducing confidence in subsequent steps. As illustrated in Figure 2, when sample I contains an initial error, self-certainty assigns lower confidence to all following steps despite their correctness, while negative perplexity fails to distinguish between reasoning paths following correct versus incorrect premises. Additionally, distributional confidence detects correct reasoning from the first few tokens.

## 4 Self-Certainty with Voting Method

While self-certainty demonstrates greater robustness than alternative confidence measures, it remains vulnerable to distortion from samples with artificially high confidence scores. Our analysis reveals that self-certainty-driven Best-of-N selection underperforms compared to self-consistency in accuracy on mathematical datasets with definitive answers when using identical N values (Table 1). This does not, however, indicate inherent inferiority. Self-consistency operates at the response layer of LLMs,

while self-certainty aggregates information at the decoding layer. By integrating both layers, we can extract more reliable responses from multiple outputs with explicit answers.

Traditional methods of combining majority voting with score-based selection, such as summing scores across samples with identical answers, suffer from sensitivity to score scaling. Similarly, using average confidence may inadequately represent frequently sampled answers. To address these limitations, we propose a Borda count-inspired approach:

First, we rank $N$ outputs of models by confidence, obtaining a ranking $[r_1, r_2, \ldots, r_N]$. We then assign votes to these ranked outputs using the following formula:

$$v(r) = (N - r + 1)^p \tag{11}$$

where $r$ is the rank of the output ($1 \leq r \leq N$). Each valid response contributes votes to its final answer proportional to its rank. The answer accumulating the highest vote total becomes the consensus selection. When $p = 0$, Equation (11) reduces to simple majority voting. As $p$ approaches infinity, the highest-ranked output dominates, reverting to pure distributional confidence selection.

Figure 3 illustrates how Borda Voting successfully identifies the correct answer by integrating both confidence ranking and answer frequency, thereby overcoming limitations of both confidence-driven selection and self-consistency. The parameter $p$, which controls ranking influence, serves as a tunable hyperparameter discussed in Section 6.2.

## 5 Experiment Setup

We compare various confidence measures for selecting reliable reasoning responses, extending evaluation to additional datasets and exploring self-certainty with voting methods.

### 5.1 Comparison of Confidence Measures

To evaluate confidence formulations from Section 3, we select the most confident response from $N$ outputs generated by Llama-3.1-8B-Instruct [Dubey et al., 2024]. We use LiveBench-Math dataset [White et al., 2024], released post-model deployment, to mitigate potential data contamination.

We sample 64 responses (temperature=0.6, top-p=0.9) and create subsets of $N = 4, 8, 16, 32, 64$ for Best-of-N selection. All measures are evaluated on identical sample sets. Responses without extractable answers are masked. We include a FirstAns baseline that selects the first extractable answer from $N$ outputs. Evaluation uses the ZeroEval framework [Lin, 2024], with results averaged across five repetitions. All experiments are run on NVIDIA A100 GPUs.

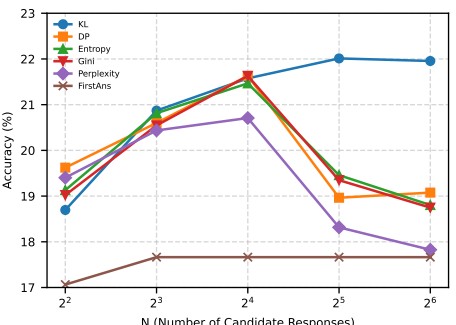

Figure 4: Best-of-$N$ selection accuracy on LiveBench-MATH across multiple confidence measures. KL achieves the best performance at larger $N$, while other measures plateau or decline after $N = 16$.

### 5.2 Validation on Additional Datasets and Combined Voting Methods

We evaluate self-certainty and Borda Voting against self-consistency, universal self-consistency (USC), greedy decoding, and FirstAns across diverse reasoning tasks. We also compare to an outcome reward model GRM-Llama3.2-3B-RewardModel-FT [Ray2333, 2024] and a process reward model (Qwen2.5-Math-PRM-7B [Zhang et al., 2025] on selected benchmarks to highlight differences between internal selection and external selection, and to quantify the resulting accuracy gaps.

The sampling strategy follows the procedures outlined in Section 5.1. For USC, we use the template from the original paper [Chen et al., 2023] (with minor wording modifications, as shown in Appendix C.2). To ensure a fair comparison, we assist USC in selecting the first valid response when it fails to choose one with an extractable answer.

We evaluate different methods using the Llama-3.1-8B-Instruct across the following benchmarks:

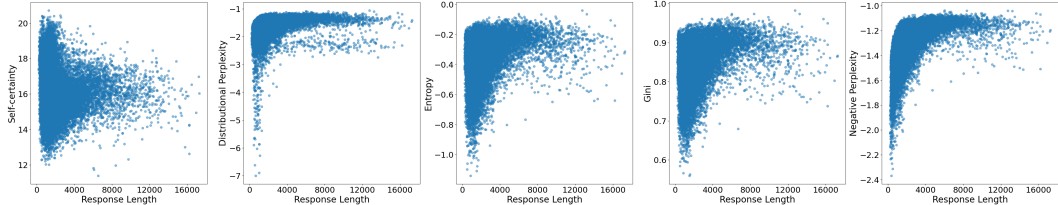

Figure 5: Scatter plot showing various confidence measures against response length (measured in number of characters) in the LiveBench-Math dataset, using the Llama-3.1-8B-Instruct model with 64 samples per question. The figure demonstrates that, with the exception of self-certainty, all other measures exhibit a bias towards longer responses.

- **Mathematical Reasoning**: We utilize the LiveBench-Math dataset [White et al., 2024], the validation set of GSM8K dataset [Cobbe et al., 2021b] and the test set of MATH dataset [Hendrycks et al., 2021].
- **Code Reasoning**: The CRUXEval-O benchmark [Gu et al., 2024] is employed, which involves predicting the output of Python codes.
- **Code Generation**: We adopt the LiveCodeBench code generation benchmark [Jain et al., 2024] to assess the improvements introduced by our methods. Note that this is an open-ended task where self-consistency cannot be applied.

For all test models and datasets, we employ Chain-of-Thought reasoning [Wei et al., 2022], except for the code generation dataset. To evaluate the generalization of our measure across different training methodologies, particularly for the recent R1-series large reasoning models [Guo et al., 2025], we test our approach on DeepSeek-R1-Distill-Llama-8B using the MATH dataset (Level 3). Given the increased reasoning time required by this model, we conduct a single trial for this experiment. To further validate and assess generalizability, we apply both USC and self-certainty to the Qwen-2.5-Coder-32B-Instruct model [Hui et al., 2024], in addition to Llama-3.1-8B-Instruct, on the LiveCodeBench dataset.

## 6 Results and Analysis

### 6.1 Self-Certainty

**KL-Divergence-Inspired Distributional Confidence Outperforms Other Measures in Best-of-N Selection.** Figure 4 shows distributional confidence measures outperform perplexity when $N \geq 16$. KL divergence uniquely continues improving as N increases to 32 and 64, demonstrating its robustness as a confidence measure with superior insight into response accuracy. Self-certainty, defined in Equation 5 as KL divergence from a uniform distribution, generalizes better than alternative empirical distributions (evaluated in Appendix B.4), confirming the efficacy of our original design.

**Self-Certainty's Robustness to Reasoning Length in Response Selection.** Figure 5 reveals a critical insight: while most confidence measures show positive correlation with response length, self-certainty remains largely invariant to reasoning length. This confirms Basu et al. [2020]'s observation that perplexity decreases with increasing output length under low $p$ values. Unlike metrics that potentially conflate verbosity with correctness, self-certainty provides an unbiased assessment of response quality, preventing models from artificially inflating confidence through extended but potentially meaningless reasoning.

**Self-Certainty Effectively Separates Correct and Incorrect Responses.** Analysis of self-certainty and negative perplexity distributions across correct, incorrect, and no-answer responses on MATH dataset Level 4 (Figure 1) demonstrates self-certainty's superior discriminative power. For self-certainty, the distributions of correct and incorrect responses are centered around distinct means, with frequencies tapering off smoothly toward both tails. In contrast, perplexity fails to distinguish between correct and incorrect responses when applied to the full dataset, despite performing adequately at small $N$ values (Figure 4). This aligns with Zhang et al. [2020]'s finding that response quality initially improves as perplexity declines but subsequently deteriorates significantly. Notably, perplexity

Table 1: Performance comparison of various methods across different datasets using Llama-3.1-8B-Instruct. Some USC results are omitted due to over 20% of the data exceeding context window limits under the settings. Self-certainty consistently outperforms sampling, greedy decoding, and perplexity, while Borda Voting with the optimal parameter $p$ delivers the best performance across all methods.

| Method | LiveBench-Math | | GSM8K | | MATH | | CRUXEval-O | | Avg. |
|---|---|---|---|---|---|---|---|---|---|
| | $N = 8$ | $N = 32$ | $N = 8$ | $N = 64$ | $N = 8$ | $N = 64$ | $N = 8$ | $N = 64$ | |
| Greedy | 12.23 | | 84.00 | | 47.96 | | 39.88 | | 46.02 |
| FirstAns | 17.66 | 17.66 | 82.08 | 82.08 | 49.08 | 49.09 | 42.93 | 42.93 | 47.94 |
| PRM | / | / | 93.48 | 95.15 | / | / | 47.53 | 48.61 | / |
| ORM | / | / | 88.57 | 89.91 | / | / | 42.00 | 39.62 | / |
| Perplexity | 20.44 | 18.32 | 87.01 | 87.81 | 53.34 | 51.96 | 44.67 | 45.10 | 51.08 |
| USC | 21.08 | - | 87.32 | 85.65 | 54.66 | - | 43.78 | 41.25 | 51.19 |
| Self-consistency | 22.50 | 26.25 | 89.42 | 90.99 | 58.60 | 63.40 | 47.58 | 50.42 | 56.15 |
| Self-certainty | 20.87 | 22.01 | 87.32 | 88.90 | 54.63 | 56.70 | 45.38 | 45.83 | 52.71 |
| - Borda ($p = 0.3$) | **23.69** | 26.47 | **89.57** | **91.07** | **59.04** | 63.60 | **47.94** | 50.42 | 56.48 |
| - Borda ($p = 0.7$) | 23.59 | 26.36 | 89.51 | 91.04 | **59.04** | 63.85 | 47.85 | 50.65 | 56.49 |
| - Borda ($p = 1.2$) | 23.21 | **26.69** | 89.51 | 90.95 | 58.86 | **64.10** | 47.93 | 50.85 | **56.51** |
| - Borda ($p = 2.0$) | 22.45 | 26.41 | 89.13 | 90.90 | 57.94 | 60.02 | 47.25 | **51.23** | 55.67 |

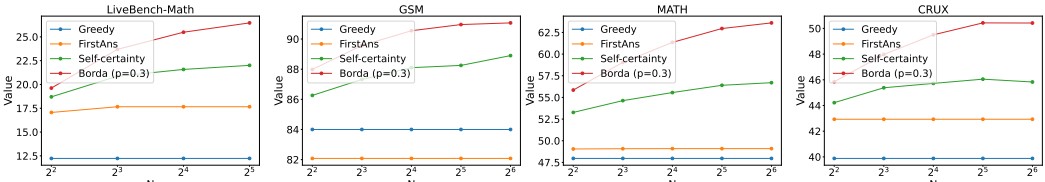

Figure 6: Performance evaluation across four datasets using different strategies with Llama-3.1-8B-Instruct. The lines show strong scaling ability of both self-certainty and Borda voting.

assigns higher confidence to no-answer responses—often resulting from self-repetition or early stopping—while self-certainty reliably assigns these responses lower confidence scores. This behavior is consistent with Basu et al. [2020]'s observation that maximizing perplexity increases self-repetition. These findings provide compelling evidence that self-certainty more effectively measures model certainty by correlating more closely with response quality.

## 6.2 Self-Certainty and Voting

**Borda Voting in Combination with Self-Certainty.** As discussed in Section 4, self-certainty can be integrated with voting methods to enhance accuracy when responses contain explicit answers. Table 2 demonstrates that self-certainty-based Borda voting outperforms majority voting, average self-certainty, and sum self-certainty on the MATH dataset.

**Performance Comparison Across Four Datasets.** Figure 6 illustrates the scaling properties of self-certainty and self-certainty-based Borda voting. Self-certainty significantly outperforms sampling, greedy decoding, and perplexity-based selection, with performance improving as $N$ increases. This confirms that self-certainty effectively measures the model's confidence in its responses, providing valuable insight into output correctness. Furthermore, Borda voting consistently outperforms self-consistency across various settings of $p$ and $N$ on all four datasets, indicating that self-certainty enhances final-answer-based voting by providing effective ranking information.

Table 2: Accuracy of different voting methods on the test set of MATH dataset using Llama-3.1-8B-Instruct. Self-certainty-based Borda voting outperforms other voting methods.

| Method | $N = 8$ | $N = 64$ |
|---|---|---|
| Majority | 58.60 | 63.40 |
| Average | 46.92 | 32.94 |
| Sum | 59.06 | 63.51 |
| Borda ($p = 0.5$) | **59.08** | 63.71 |
| Borda ($p = 1.2$) | 58.86 | **64.10** |

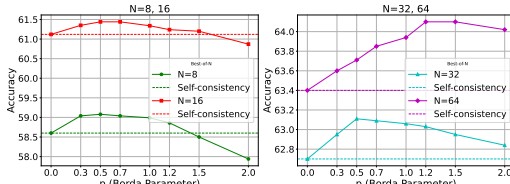
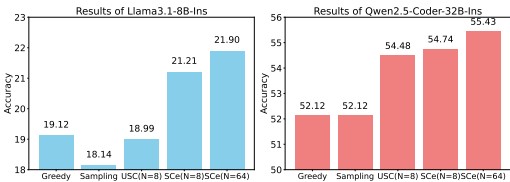

Figure 7: Performance of Borda voting on MATH dataset using Llama-3.1-8B-Instruct with varying $p$ and $N$. Accuracy initially increases with $p$, peaks, then declines. The optimal $p$ varies with $N$. Note that self-consistency corresponds to Borda voting with $p = 0$.

Figure 8: Comparison of self-certainty and USC on the LiveCodeBench code generation task. The results show that self-certainty outperforms USC and greedy decoding on both Llama-3.1-8B-Instruct and Qwen-2.5-Coder-32B-Ins models, with performance improving as $N$ increases.

**Optimizing the Borda Parameter $p$ for Different $N$.** Figure 7 shows the relationship between the Borda parameter $p$ in Equation 11 and selection efficiency across varying sample sizes $N$. The optimal $p$ increases from 0.5 to 1.2 as $N$ increases from 8 to 64, suggesting that larger sample sizes require stronger control from self-certainty. For practical applications, grid search remains the most effective approach for determining the optimal $p$, though a simple heuristic is to use $p = 0.3$ when $N \leq 16$ and $p = 1.2$ when $N \geq 32$, with the understanding that the optimum is model- and task-dependent. Tuning $p$ on a small portion of the task-specific validation set is recommended to achieve the best performance, as convergence of optimal $p$ typically occurs within a few hundred evaluation queries.

## 6.3 Generalization

**Generalization of Self-Certainty on Open-Ended Generation Tasks.** Self-consistency faces limitations with creative, open-ended tasks like code generation, where each sample produces unique answers, defaulting to standard sampling. Both USC and our self-certainty method address this limitation. Comparing these approaches on LiveCodeBench (Figure 8), we find that USC underperforms greedy decoding on Llama-3.1-8B-Instruct, likely due to limited consistency recognition capabilities. This is confirmed by results from the larger Qwen model, where USC successfully outperforms greedy decoding. In contrast, self-certainty consistently outperforms greedy decoding across both models and surpasses USC on Qwen-2.5-Coder-32B-Ins, with performance scaling positively with sample size $N$.

**Generalization of Self-Certainty on Reasoning Models.** Recent work on DeepSeek-R1 [Guo et al., 2025] shows that reinforcement learning with verifiable rewards and long-chain-of-thought (CoT) significantly enhance LLM reasoning capabilities. Our evaluation of self-certainty on DeepSeek-R1-Distill-Llama-8B (Table 3) demonstrates that it consistently outperforms both greedy decoding and sampling, with performance scaling with $N$. Additionally, Borda voting with self-certainty surpasses self-consistency when using appropriate $p$ values. These results confirm the robustness of our methods across various fine-tuning approaches.

Table 3: Accuracy of various methods on the Level 3 test set of the MATH dataset using DeepSeek-R1-Distill-Llama-8B (single trial). Self-certainty outperforms Greedy and FirstAns, while Borda Voting with an appropriate $p$ surpasses self-consistency.

| Method | $N = 4$ | $N = 16$ | $N = 64$ |
|---|---|---|---|
| Greedy | 77.54 | 77.54 | 77.54 |
| FirstAns | 81.17 | 81.43 | 81.43 |
| Self-consistency | 83.64 | 86.47 | 87.62 |
| Self-certainty | 83.29 | 83.73 | 84.08 |
| - Borda ($p = 0.3$) | 84.79 | 87.00 | 87.80 |
| - Borda ($p = 0.7$) | 84.70 | 86.91 | 87.62 |
| - Borda ($p = 1.2$) | 84.62 | 87.00 | 88.06 |
| - Borda ($p = 2.0$) | 83.29 | 87.00 | 87.98 |

## 7 Discussion and Future Research

Our study establishes self-certainty as a scalable, lightweight, and effective metric for evaluating LLM outputs, particularly for open-ended and complex reasoning tasks. While it scales well with increasing sample size and outperforms existing reward-free methods across multiple settings, several directions for refinement remain.

First, self-certainty can underperform self-consistency on problems with definitive, convergent answers (Section 6). This reflects the complementary nature of different aggregation methods rather than a limitation. Combining self-certainty with answer-level voting mechanisms—such as Borda voting—bridges this performance gap, achieving results that rival or exceed self-consistency. These findings suggest that self-certainty could enhance reward model design by shifting from token-level scoring to distribution-aware confidence estimation. The use of KL divergence from a uniform distribution offers greater robustness than traditional average log-probability metrics and may lead to more stable reward training objectives. In practical terms, the metric can substitute for selected-token probabilities in applications such as soft self-consistency [Wang et al., 2024b], with potential performance gains.

Second, our implementation uses a simple averaging strategy for aggregating token-level confidence (Equation 4) and a basic power function for distributing votes in Borda voting (Equation 11). Future work should explore more sophisticated aggregation functions or data-driven approaches for learning optimal vote weighting schemes to improve accuracy in specialized applications.

Self-certainty also enables broader research opportunities. Its computational efficiency makes it ideal for test-time optimization techniques [Snell et al., 2024], producing higher-quality outputs without additional inference passes. It offers potential value in data filtering, auto-labeling, and reinforcement learning pipelines [Bai et al., 2022, Ouyang et al., 2022], where confidence estimation is crucial. Specifically, self-certainty could guide reward shaping or provide intrinsic signals for autonomous agents, better aligning learning objectives with model certainty.

# 8 Conclusion

In this paper, we introduce self-certainty and self-certainty-based Borda voting as novel approaches for evaluating and enhancing model response performance. Self-certainty functions as an internal measure of response quality, demonstrating robustness in several key aspects. Compared to traditional scoring methods, such as average log probability and perplexity, it offers superior scalability when applied to Best-of-N selection. Additionally, the ranking information provided by self-certainty improves chain-of-thought reasoning and outperforms universal self-consistency (USC) in code generation tasks. Its stability, flexibility, and generalizability make it applicable across a wide range of domains, with the potential to enhance the autonomous learning capabilities of LLMs.

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

# A  Theoretical Analysis

## A.1  Log-likelihood Ascent of Target Token Increases Self-Certainty

**Setup and definition.**  For a single decoding step, let $p = (p_1, \ldots, p_V) \in \Delta^{V-1}$ denote the next-token distribution over a vocabulary of size $V$ with logits $z \in \mathbb{R}^V$ and $p = \mathrm{softmax}(z)$. The token-wise self-certainty is

$$\mathrm{SC}(p) = -\frac{1}{V} \sum_{j=1}^{V} \log(V\, p_j),$$

**Differentiate Self-Certainty.**  Let $\delta_{jk}$ be the Kronecker delta. Two standard derivatives are

$$\frac{\partial p_j}{\partial z_k} = p_j(\delta_{jk} - p_k), \qquad \frac{\partial\, \mathrm{SC}}{\partial p_j} = -\frac{1}{V} \cdot \frac{1}{p_j}.$$

By the chain rule,

$$\frac{\partial\, \mathrm{SC}}{\partial z_k} = \sum_{j=1}^{V} \frac{\partial\, \mathrm{SC}}{\partial p_j} \frac{\partial p_j}{\partial z_k} = -\frac{1}{V} \sum_{j=1}^{V} \frac{1}{p_j} p_j(\delta_{jk} - p_k) = -\frac{1}{V}(1 - V p_k) = p_k - \frac{1}{V}. \tag{1}$$

**Update that Increases a Specific Token's Probability.**  Fix a target token index $y^*$. A gradient-*ascent* step on $\log p_{y^*}$ moves logits along

$$\Delta z_k \ \propto\ \frac{\partial \log p_{y^*}}{\partial z_k} = \delta_{k,y^*} - p_k, \tag{2}$$

so for step size $\eta > 0$ we consider $z(\eta) = z + \eta\, \Delta z$ and $p(\eta) = \mathrm{softmax}(z(\eta))$.

**Directional Change of Self-Certainty.**  Using (1) (2), the directional derivative of SC at $\eta = 0$ is

$$\frac{d}{d\eta} \mathrm{SC}(p(\eta)) \Big|_{\eta=0} = \sum_{k=1}^{V} \frac{\partial\, \mathrm{SC}}{\partial z_k} \Delta z_k = \sum_{k=1}^{V} \left(p_k - \frac{1}{V}\right)(\delta_{k,y^*} - p_k) = p_{y^*} - \sum_{k=1}^{V} p_k^2. \tag{3}$$

Let $\|p\|_2^2 := \sum_k p_k^2$. Then (3) gives the exact criterion

$$\frac{d}{d\eta} \mathrm{SC}(p(\eta)) \Big|_{\eta=0} > 0 \iff p_{y^*} > \|p\|_2^2. \tag{4}$$

**Theorem A.1** (When self-certainty increases under log-likelihood ascent)**.**  *Under the self-certainty* $\mathrm{SC}(p) = \mathrm{KL}(U\|p)$*, a (stochastic) gradient-ascent step on* $\log p_{y^*}$ *increases the token-position's self-certainty to first order in the step size if and only if* $p_{y^*} > \|p\|_2^2$*.*

*Proof.*  Immediate from (3)–(4). □

**Corollary A.2** (Argmax case)**.**  *If* $y^* = \arg\max_k p_k$ *and* $p$ *is not one-hot, then* $p_{y^*} > \|p\|_2^2$ *and thus the ascent step strictly increases* SC*. Indeed,* $\|p\|_2^2 \le \max_k p_k$ *with equality only when* $p$ *is one-hot.*

**Remarks.**  (i) If $p_{y^*}$ starts below $\|p\|_2^2$, the first few ascent steps can decrease self-certainty. Once $p_{y^*}$ exceeds $\|p\|_2^2$, further ascent increases it by Theorem A.1. (ii) Sentence-level self-certainty inherits the same monotonicity condition token-wize.

# B  More Experiment Results

## B.1  Oracle Best-of-N Selection Performance and Scaling Effects on LiveCodeBench

In our experiment described in Section 5.2, we evaluate the performance of Llama-3.1-8B-Instruct and compare Borda voting and self-certainty against the upper bound of Best-of-N selection methods, as shown in Figure 9. While both methods demonstrate continued improvement as $N$ increases, they remain significantly outperformed by the Oracle selection method, which assumes perfect knowledge of the correct answer.

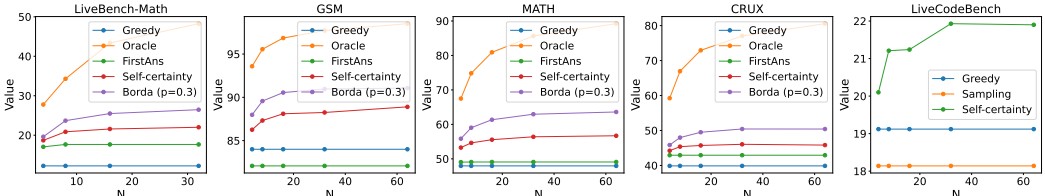

Figure 9: Performance across five datasets using different strategies with Llama-3.1-8B-Instruct. The oracle selection method significantly outperforms the other strategies. Additionally, both Borda voting and self-certainty demonstrate strong scaling effects.

## B.2 Average Self-Certainty Across Difficulty Levels on MATH Dataset

To explore how self-certainty is influenced by question difficulty, we evaluate the average self-certainty score across different difficulty levels of the MATH dataset, as shown in Figure 10. The results indicate that the average self-certainty generally decreases as the difficulty level increases, regardless the correctness of the questions. This trend makes self-certainty a promising parameter-free approach for assessing question difficulty, offering a potential alternative to training classifiers [Snell et al., 2024] when determining difficulty levels for scaling test-time compute strategies.

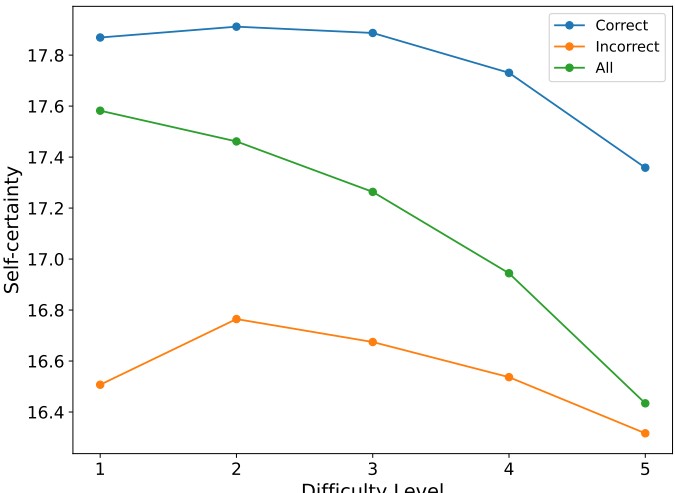

Figure 10: Comparison of the average self-certainty score on the MATH test dataset across increasing difficulty levels using Llama-3.1-8B-Instruct for 64 responses per question (single trial). The average self-certainty decreases as questions become more challenging. This trend is observed for both correct and incorrect responses.

## B.3 Evaluation of Methods Across Difficulty Levels on the MATH Dataset

We evaluate different methods across varying difficulty levels of reasoning problems. Figure 11 presents the performance of various methods on the MATH dataset at different difficulty levels. As question difficulty increases, the scaling effect of Borda voting and self-certainty becomes more pronounced, demonstrating their effectiveness in handling more challenging reasoning tasks.

## B.4 Replacing Uniform Distribution with Empirical Distribution

In Equation 5, we define tokenwise self-certainty as the KL divergence between the generated token distribution and a uniform distribution, which quantifies deviation from random sampling. An alternative approach replaces the uniform distribution with an empirical token distribution estimated from training data. To evaluate the impact of this modification, we conduct the following experiment.

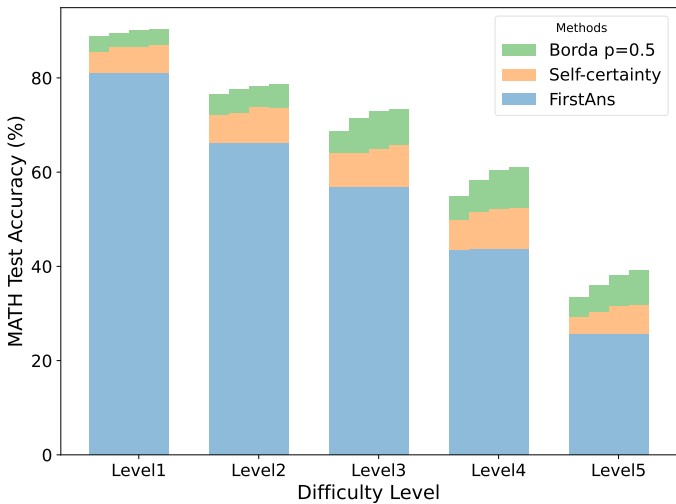

Figure 11: Comparison of evaluation methods on the MATH test dataset across increasing difficulty levels using Llama-3.1-8B-Instruct. The four bars in each difficulty bin correspond to an increasing choice of N in the Best-of-N selection (8, 16, 32, and 64 generations). Performance differences among settings become more pronounced as the difficulty level increases.

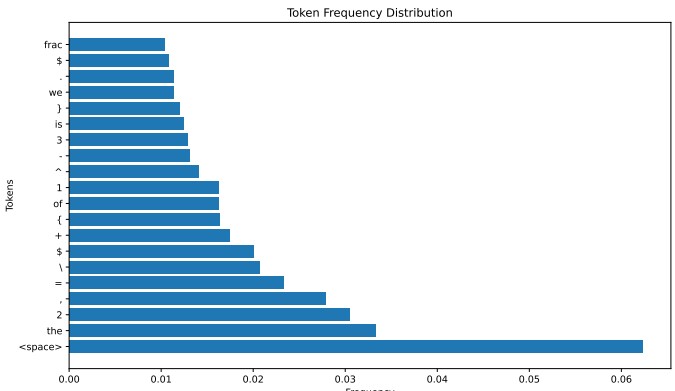

Figure 12: Frequency of the top 20 tokens in Llama-3.1-8B-Instruct responses to MATH training set questions (generated eight times per question).

We first estimate token frequencies in the MATH training set by generating eight responses per question and averaging token occurrences. The resulting empirical distribution is approximated from these frequencies, with the 20 most frequent tokens shown in Figure 12. We then compute KL divergence between the generated token distribution and the empirical distribution, using this as the self-certainty measure for Best-of-N selection. This experiment was conducted for a single trial, with all other setup parameters as described in Section 5.1.

Results show that replacing the uniform distribution with the empirical distribution has minimal impact on MATH test accuracy but leads to a noticeable performance drop on GSM8K, suggesting a sensitivity to distributional shifts. Thus, we recommend retaining the uniform distribution in Equation 5 for improved generalization.

## B.5  Comparison of Voting Methods with Different Confidence Metrics

We further compare the voting method using self-certainty with variants that rely on other confidence metrics. The results, presented in Table 5, show that for large values of $N$, self-certainty consistently

Table 4: Accuracy of various self-certainty definitions for Best-of-N selection on the MATH and GSM8K test sets using Llama-3.1-8B-Instruct (single trial). The empirical distribution is derived by sampling from the MATH training dataset. While the empirical self-certainty results are comparable to those based on a uniform distribution for the MATH test set, it is significantly outperformed by the latter, likely due to a distributional shift.

| Base Distribution | MATH | | GSM8K | |
|---|---|---|---|---|
| | $N = 8$ | $N = 64$ | $N = 8$ | $N = 64$ |
| Uniform | 54.60 | 56.46 | 87.19 | 88.55 |
| Empirical | 54.70 | 56.78 | 85.97 | 86.35 |

outperforms the alternatives when the Borda exponent $p$ is properly tuned. This highlights the robustness of self-certainty relative to other metrics.

Table 5: Accuracy of Borda voting methods on the test set of MATH-level5 test set (single trial) using Llama-3.1-8B-Instruct.

| Method | Perplexity | Self-certainty | (D) Entropy | (D) Perplexity | (D) Gini |
|---|---|---|---|---|---|
| Self-consistency | 37.99 | 37.99 | 37.99 | 37.99 | 37.99 |
| Borda ($p = 0.5$) | 38.60 | 38.90 | 39.12 | 38.14 | 39.35 |
| Borda ($p = 1.2$) | 38.60 | 39.43 | 38.40 | 38.14 | 38.75 |

## B.6 Comparison with Normalized Weighted Sum from Self-Consistency

The normalized weighted sum (NWS) variant of self-consistency [Wang et al., 2022] combines both voting and model logit information by summing the sentence-level average log probabilities of groups of completions that yield the same final answer. However, the authors show that incorporating sample token probabilities in this manner generally does not improve the standard voting-based self-consistency and sometimes underperforms the vanilla method.

For completeness, we compare our Borda voting approach with the normalized weighted sum version of self-consistency on the MATH-Level5 test set using Llama-3.1-8B-Instruct. As shown in Table 6, our method achieves higher accuracy, further validating the effectiveness of both the self-certainty metric and the Borda voting framework.

Table 6: Comparison between self-consistency (normalized weighted sum) and standard self-consistency on MATH-Level5 with Llama-3.1-8B-Instruct using $N = 64$ samples (single trial). Borda voting with an appropriate exponent $p$ outperforms the normalized weighted sum variant.

| | Normalized-Probability | + Weighted-Sum | Self-Certainty | + Borda ($p$=2.0) |
|---|---|---|---|---|
| **Accuracy** | 27.34 | 38.75 | 31.42 | 39.43 |

## C Case Study and Examples

In this section, we present several examples to illustrate the practical behavior of self-certainty and universal self-consistency (USC). These examples are drawn from Llama-3.1-8B-Instruct generating responses to questions from the GSM8K and LiveCodeBench code generation datasets.

### C.1 Self-Certainty

In this example, self-certainty correctly assigns higher confidence to the correct answer. Notably, in Response 0, the model does not fully adhere to the instruction to enclose all reasoning within the specified JSON field. Instead, it follows its trained reasoning process first and then summarizes the reasoning path in the `reason` field. This approach appears to be a compromise between the model's training objective and the given instruction, balancing structured formatting with its natural reasoning process.

## Self-Certainty Example

USER:
## Question:
Kylar went to the store to buy glasses for his new apartment. One glass costs $5, but every second glass costs only 60% of the price. Kylar wants to buy 16 glasses. How much does he need to pay for them?
## Instruction:
Please answer this question by first reasoning and then providing your answer.
Present your reasoning and solution in the following json format. Please show your final answer in the answer field, e.g.,"answer": "42". 'json{ "reasoning": "___", "answer": "___"}

Response 0:
ASSISTANT:
##Step 1: Calculate the cost of the first glass.
The cost of the first glass is $5.
##Step 2: Calculate the cost of every second glass.
Every second glass costs 60% of the price of the first glass, which is $0.6 \times 5 = 3$ dollars.
##Step 3: Determine the number of glasses that are full price and the number of glasses that are discounted.
Kylar wants to buy 16 glasses. Since every second glass is discounted, we can divide 16 by 2 to get the number of full-price glasses and the number of discounted glasses. $\frac{16}{2} = 8$ full-price glasses and 8 discounted glasses.
##Step 4: Calculate the total cost of the full-price glasses.
8 full-price glasses at $5 each is $8 \times 5 = 40$ dollars.
##Step 5: Calculate the total cost of the discounted glasses.
8 discounted glasses at $3 each is $8 \times 3 = 24$ dollars.
##Step 6: Calculate the total cost of all the glasses.
The total cost is the sum of the cost of the full-price glasses and the cost of the discounted glasses. $40 + $24 = $64.
##Step 7: Determine the total amount Kylar needs to pay for the glasses.
The total amount Kylar needs to pay is $64.
The final answer is: {"reasoning": "Kylar wants to buy 16 glasses. The first glass costs $5, and every second glass costs 60% of the price of the first glass, which is $3. He buys 8 full-price glasses at $5 each and 8 discounted glasses at $3 each. The total cost of the full-price glasses is $40, and the total cost of the discounted glasses is $24. The total cost is $40 + $24 = $64.", "answer": "64"}
**(Self-certainty: 17.81)**

Response 1:
ASSISTANT:
{"reasoning": "To find the total cost, we need to calculate the cost of the first glass and the cost of the remaining glasses. The first glass costs $5. The second glass costs 60% of $5, which is 0.6 * 5 = $3. The third glass costs 60% of $3, which is 0.6 * 3 = $1.80. We can see that the price of each subsequent glass is 60% of the previous glass. So, the price of the glasses follows a geometric sequence: 5, 3, 1.80, 1.08, . . . . To find the cost of 16 glasses, we can use the formula for the sum of a geometric series: $S = a * (1 - r^n)/(1 - r)$, where a is the first term, r is the common ratio, and n is the number of terms. In this case, a = 5, r = 0.6, and n = 16. Plugging in the values, we get $S = 5 * (1 - 0.6^{16})/(1 - 0.6)$. Calculating the value of $0.6^{16}$, we get 0.0000655. Now, we can plug this value back into the formula: $S = 5 * (1 - 0.0000655)/(1 - 0.6) = 5 * 0.9999345/0.4 = 5 * 2.499335 = 12.496675$. Rounding to two decimal places, the total cost is $12.50.", "answer": "12.50"}
**(Self-certainty: 17.13)**

## C.2 Universal Self-Consistency

The first example is taken from the GSM8K dataset, while the second comes from the LiveCodeBench dataset. The prompt has been adapted from Chen et al. [2023]. We observe that in Example 1, the model evaluates the answers step by step to assess consistency. In contrast, the model jumps directly to the conclusion in Example 2. This difference may stem from the clarity of the task: in a MATH problem with a definite answer, the model can logically assess each step of the reasoning process, as seen in Example 1. However, for the LiveCodeBench example, where the task involves determining the most consistent code, the model struggles to elaborate on the reasoning process and relies on direct reasoning to arrive at a conclusion.

---

**USC Prompt Example 1**

USER:
I have generated the following responses to the question: Elise is learning to write and decides to keep re-writing the alphabet until she knows it. She writes it in full twice, writes half of it once, then re-writes everything she has already written. How many letters has Elise written in total?

Response 0: {"reasoning": "To find the total number of letters Elise has written, we first need to determine the number of letters in the alphabet. There are 26 letters in the English alphabet. Elise writes it in full twice, which means she writes 26 letters twice, so 26 * 2 = 52 letters. Then, she writes half of it once, which is 26 / 2 = 13 letters. So, the total number of letters written so far is 52 + 13 = 65. Finally, she re-writes everything she has already written, which is 65 letters. Therefore, the total number of letters Elise has written is 65 + 65 = 130.", "answer": "130"}

. . .

Response 3: {"reasoning": "To solve this problem, we need to calculate the total number of letters Elise has written. First, she writes the alphabet twice, which has 26 letters. So, she has written 26 + 26 = 52 letters. Then, she writes half of the alphabet once, which is 13 letters. So, she adds 13 more letters to the total. Now, she re-writes everything she has already written, which means she re-writes 52 + 13 = 65 letters. To find the total number of letters Elise has written, we need to add the number of letters she wrote initially (52) and the number of letters she re-wrote (65). The total number of letters is 52 + 65 = 117.", "answer": "117"}

. . .

Evaluate these responses. Select the most consistent response based on majority consensus. Format your answer as "The most consistent response is Response X" (without quotes).

ASSISTANT:
To evaluate the responses, I will analyze the answers and reasoning provided by each response.Response 0 and Response 1 have the same answer (130) and similar reasoning. · · ·.The most consistent response is Response 0.

---

**USC Prompt Example 2**

USER:
I have generated the following responses to the question: Slavic is preparing a present for a friend's birthday. He has an array $a$ of $n$ digits, and the present will be the product of all these digits. Because Slavic is a good kid who wants to make the biggest product possible, he wants to add 1 to exactly one of his digits.
**What is the maximum product Slavic can make?**
**Input:**

---

- The first line contains a single integer $t$ $(1 \leq t \leq 10^4)$ — the number of test cases.
- The first line of each test case contains a single integer $n$ $(1 \leq n \leq 9)$ — the number of digits.
- The second line of each test case contains $n$ space-separated integers $a_i$ $(0 \leq a_i \leq 9)$ — the digits in the array.

**Output:**

- For each test case, output a single integer — the maximum product Slavic can make, by adding 1 to exactly one of his digits.

Sample Input: $\cdots$
Sample Output: $\cdots$

**Response 0:**

```python
# YOUR CODE HERE
import sys

def max_product(n, arr):
    min_digit = min(arr)
    max_product = 1
    for i in range(n):
        if arr[i] == min_digit:
            arr[i] += 1
            break
    for num in arr:
        max_product *= num
    return max_product

t = int(input())
for _ in range(t):
    n = int(input())
    arr = list(map(int, input().split()))
    print(max_product(n, arr))
```

This code first finds the minimum digit in the array and increments it by 1. Then, it calculates the product of all digits in the array. The time complexity is O(n), where n is the number of digits in the array.
$\cdots$

Evaluate these responses. Select the most consistent response based on majority consensus. Format your answer as "The most consistent response is Response X" (without quotes).

ASSISTANT:
The most consistent response is Response 0.

