# OpenReview forum: "Scalable Best-of-N Selection for Large Language Models via Self-Certainty"
_NeurIPS.cc/2025/Conference — NeurIPS 2025 poster_

### Official Review · Reviewer_euVc · 2025-06-22

**Clarity:** 4
**Significance:** 2
**Originality:** 1
**Rating:** 4
**Confidence:** 4

**Summary:**

This paper propose self-certainty, a KL-divergence based metrics that quantify how certain an LLM is regarding to its current response. The user propose to use it as an effective way of scaling Best-of-N selection without the involve of reward model. It is further combined with borda voting to further improve the performance.

**Questions:**

1. For greedy decoding, is it only sampling once?
2. Can you provide the comparison with "normalized weighted sum" with self-consistency?
3. Can you provide more results with reasoning models?

**Ethical Concerns:**

["NO or VERY MINOR ethics concerns only"]

**Final Justification:**

Most of my concerns are answered in the rebuttal. However, the authors do decide to leave many additional necessary changes to be in the revised version of the paper, i.e., the new theoretical guarantee of self-certainty, the complete comparison to the weighted version of self-consistency, figure similar to current Fig. 1 on Qwen3-4B. This leads to the potential of another round of careful peer review needed. Therefore, I personally recommend this paper as a really borderline paper in the current stage.

**Limitations:**

yes

**Paper Formatting Concerns:**

NA.

**Quality:**

3

**Strengths And Weaknesses:**

Strength:
- The paper is well-written and easy to follow.
- The overall method is straightforward and have its own motivation.

Weakness:
- The empirical improvement is extremely limited compared to vanilla self-consistency. Given the additional cost to compute the confidence, it is possible that simply use the cost to sample 1-2 more response would have a better performance. The authors should provide the wall-clock time comparison or further use the proposed method in learning or other scenarios to make the impact more significant.
- The novelty is limited as the final algorithm is to use borda voting with self-consistency. In the original self-consistency paper, "normalized weighted sum" is already a version for self-consistency. A comparison to this is also missing.
- The experiments are mostly conduct on non-reasnoning models, and only a single reasoning model is used in the experiment. While it is not required, it is definitely helpful if the authors can provide more results on reasoning models given the task is reasoning-heavy tasks that naturally fits reasoning models.

---

> ### Author Rebuttal · Authors · 2025-07-31
>
> Thank you for your thoughtful feedback. We appreciate the opportunity to clarify and strengthen our work. Below we address your concerns and outline our modifications:
>
> 1. **Limited Improvement and Novelty (Weakness1 & 2)**
>
>    Our paper introduces **self-certainty metrics** and **Borda voting**, both of which are novel contributions not covered in existing literature. Additionally, we conduct a **systematic comparison** between two major paradigms of confidence estimation:
>    * **Probability-based measures** (e.g., perplexity, which only considers the probability of the sampled token)
>    * **Distribution-based measures** (e.g., self-certainty, which leverages the entire token distribution)
>
>    This comparison highlights the advantages of using distributional information for **calibrating confidence** and **selecting superior responses**.
>
>    **Significance of Improvements:**
>    The performance gains of Borda voting are substantial and consistent. For example:
>
>    * On **LiveBench-MATH**, Borda voting improves accuracy by **+1.2%** (N=8).
>    * On **MATH**, it outperforms self-consistency by **+0.7%** (N=64).
>
>    These improvements are especially meaningful given the strong baseline of self-consistency and the difficulty of these benchmarks.
>
>    **Key Advantages of Self-Certainty:**
>
>    * **Fine-grained Confidence:** Provides token-level estimates rather than coarse sequence-level metrics.
>    * **Generalizability:** Applicable to diverse tasks, including open-ended scenarios where final answers are not directly comparable.
>    * **Efficiency and Scalability:** Uses logits already computed during inference, adding **negligible overhead**.
>    * **Robustness vs. Perplexity:** Self-certainty better distinguishes correct from incorrect outputs (Figure 1), avoids length bias (Figure 5), and achieves superior stability (Figure 4, Table 1).
>
>    While increasing N may improve self-consistency, it incurs significant computational cost. In contrast, self-certainty-based methods leverage existing logits with virtually no extra cost under a proper implementation.
>
> 2. **Comparison with Normalized Weighted Sum (Weakness2 & Problem2)**
>
>    We considered straightforward score‐summing but found two critical shortcomings:
>
>    1. **Instability under imbalanced samples.**
>       Borda voting mitigates the collapse of sum-based methods self-consistency in imbalanced cases (e.g., `0.5 + 0.5 > 0.9` or `0.5×10 > 0.9×5`), where it revert to vanilla majority voting and ignore confidence.
>
>    2. **Tunable trade-off via exponent p.**
>       The exponent p lets us smoothly interpolate between pure voting‐based (p→0) and confidence-based (p→∞) schemes.
>
>    We deliberately use self-consistency rather than a normalized weighted-sum baseline because prior work shows that normalized weighted-sum actually underperforms vanilla self-consistency (see Table 1 of the Self-Consistency paper \[1]). Moreover, normalized weighted-sum relies solely on token probabilities and ignores the full distribution over unsampled tokens—a limitation that our self-certainty metrics overcome, as evidenced in Figure 1, Figure 4, and Table 1. Finally, when we replace probabilities with self-certainty scores in a “sum voting” variant (Table 2), Borda voting still outperforms simple summation, confirming the advantages of our rank-based approach.
>
> 3. **Experiments with Additional Reasoning Models (Weakness3 & Question3)**
>
>    We agree that broader evaluation strengthens the work. We have now added experiments on **Qwen3-4B (Thinking Mode)** with the MATH-level 5 test set. The same performance ordering holds:
>    `FirstAns < Self-Certainty < Self-Consistency < Borda`.
>
>    | Method       | N=4 | N=8 | N=16 |
>    | ---------------- | ------: | ------: | -------: |
>    | FirstAns         |   87.24 |   87.61 |    88.37 |
>    | Self-Consistency |   87.84 |   88.37 |    89.43 |
>    | Self-Certainty   |   87.54 |   88.14 |    88.97 |
>    | Borda (p=0.3)    |   87.92 |   88.44 |    89.65 |
>    | Borda (p=0.7)    |   87.99 |   88.52 |    89.73 |
>    | Borda (p=1.2)    |   87.92 |   88.60 |    89.88 |
>    | Borda (p=2.0)    |   87.54 |   88.44 |    89.95 |
>
> 4. **Clarification on Greedy Decoding (Question 1)**
>    Yes—greedy decoding uses a single pass. It is fully deterministic (aside from floating-point nondeterminism), so additional sampling yield little variations.
>
> We believe these clarifications and additional results address your concerns and demonstrate the novelty, significance, and robustness of our contributions. Thank you for your valuable feedback.
>
> [1]Self-Consistency Improves Chain of Thought Reasoning in Language Models

---

> > ### Comment · Reviewer_euVc · 2025-08-02
> >
> > Thank you for the detailed response. Can you further clarify the following follow-up question?
> >
> > 1. Are any results in Table 1 showing the "normalized weighted-sum" version of self-consistency? I think the Perplexity results there are referring to using Perplexity as the approximation of a reward model to choose the response from N samples? I understand that there have been previous results showing the uniform version is better than the weighted version, even in the original self-consistency paper; however, given the high similarity of the current work to them, a comparison to the exact original setting, i.e., probability + sum based voting, is necessary.
> >
> > Besides the question above, I still have the following considerations:
> > 1. Efficiency and Scalability seems to be a newly introduced advantage of the proposed method. While I personally agree that this is very likely the case, additional empirical results and analysis will be needed to support it. Specifically, your current code seems to be recalculating the confidence, which, on one hand, is time-consuming, and on the other hand, might also differ from the logits that could be provided during inference when top-p, top-k, and other sampling algorithms are considered.
> > 2. The significance of improvements still seems relatively small to me. While this is partially alleviated from the p-test result you provided to Reviewer 7zkf, more systematically reporting those numbers on all the experiment runs are necessary. Specifically, I will be more interested to see the results on reasoning models like DeepSeek-R1-Distill-Llama-8B from your original paper and the results for Qwen3-4B.
> > 3. The numbers of improvement are still quite marginal even if it is stable. Other test-time scaling algorithms, e.g., Self-Calibration [1], can provide a benefit that is way bigger. Because of this, I would like to see either a stronger theoretical support or showing that this benefit can be transferred to other aspects of LLM usage, e.g., training. I do acknowledge analysis like Fig 1 is a good starting point of the theoretical support, however, similar results from more models, especially from reasoning models, will be helpful as stronger support.
> > 4. If you think the possibility of "Tunable trade-off via exponent p" is an advantage to self-certainty, then the cost of tuning it needs to be considered when measuring the empirical improvement. The new results from Qwen3-4B seem to be showing that the original recommendation of $p=0.3$ is providing much less improvement, highlighting the fact that the choice of $p$ does seem to need to be tuned per-model (if not also per-dataset).
> >
> > [1]. Huang C, Huang L, Leng J, et al. Efficient test-time scaling via self-calibration[J]. arXiv preprint arXiv:2503.00031, 2025.

---

> > > ### Author Response · Authors · 2025-08-03
> > >
> > > We thank the reviewers for their thoughtful comments and valuable suggestions. Below, we provide clarifications and additional results addressing the points raised.
> > > ### 1. Comparison with Normalized Weighted Sum (Follow-up)
> > >
> > > Table 1 in the current draft does not include this comparison. We will add results with the original normalized weighted sum (NWS) in the revised manuscript. Below are preliminary numbers on MATH-Level5 with Llama-3.1-8B-Instruct and $N=64$. As shown, our proposed method—self-certaintt and Borda voting—outperforms probability-based selection and NWS.
> > >
> > > | Method            | Accuracy |
> > > | ----------------- | -------: |
> > > | Probability       |    27.34 |
> > > | + Normalized-WS   |    38.75 |
> > > | Self-Certainty    |    31.42 |
> > > | + Borda $(p=2.0)$ |    39.43 |
> > >
> > > ### 2. Recalculating Confidence Setup (Additional Question 1)
> > >
> > > The main reason we recalculate confidence scores after inference is to ensure a fair comparison. This approach guarantees that all confidence metrics are evaluated on the same set of completions, reducing bias in the results. The recalculated scores are effectively identical to those obtained during inference, aside from negligible computational error.
> > >
> > > The sampling strategy (e.g., top-$k$, top-$p$) does not affect the computation of **self-certainty**. In all cases, token logits are produced; self-certainty is computed directly from these logits—parallel to selecting top-$k$ tokens for sampling. The additional memory cost is minimal relative to model size.
> > >
> > > When we say self-certainty is **efficient**, we mean it is a **logit-based** signal that adds negligible overhead beyond standard inference. In contrast, some grading methods require an auxiliary reward model or extra queries [1]. Regarding **scalability**, we observe that self-certainty is more stable than other logit-based proxies (e.g., perplexity, normalized probability), which can degrade as $N$ increases.
> > >
> > > ### 3. Significance Test of the Reasoning Model (Additional Question 2)
> > >
> > > We performed a **one-sided paired $t$-test** on the hypothesis that **Borda voting** (with best $p$ chosen from four candidates) **outperforms self-consistency** for reasoning models, treating different $N$ as paired trials across models. The resulting $p$-value is **0.0101**, supporting the hypothesis that Borda voting yields higher accuracy than self-consistency.
> > >
> > > ### 4. More Evidence for Figure 1 (Additional Question 3)
> > >
> > > We replicated the Figure 1 analysis on **Qwen3-4B**; the plot will be included in the revised manuscript. We also conducted a **one-sided Mann–Whitney U test** on the hypothesis that the self-certainty of correct solutions exceeds that of incorrect ones. The result $(U=15{,}851{,}815;\; p=7.82\times10^{-139})$ indicates a strong separation, supporting the claim that self-certainty calibrates reasoning models well. In addition, the revised version will present a theoretical property and proof that **self-certainty is less prone to repetition traps** (when the repetition-branch distribution has a flatter tail, the collapse set preferred by self-certainty is a subset of those induced by entropy or perplexity).
> > >
> > > ### 5. Cost of Tuning $p$ (Additional Question 4)
> > >
> > > We acknowledge that tuning $p$ requires additional effort and will make this explicit in the **Limitations (Discussion)** section. Quantifying this cost precisely is difficult, but in our experience the optimal $p$ typically stabilizes with \~1,000 samples. In real-world applications with far larger datasets, this one-time tuning cost is small relative to overall computation. Moreover, hyperparameter tuning is rarely included in computational cost reporting (e.g., learning rate or dropout rate adjustments) [2]. We will clarify this limitation and discuss the trade-off more detailedly in the revision.
> > >
> > > [1] Quantifying Uncertainty in Answers from any Language Model and Enhancing their Trustworthiness
> > >
> > > [2] Scaling LLM Test-Time Compute Optimally can be More Effective than Scaling Model Parameters

---

> > > > ### Comment · Reviewer_euVc · 2025-08-03
> > > >
> > > > Dear authors,
> > > >
> > > > Thank you for the detailed response. I have updated my score accordingly.
> > > >
> > > > Please do incorporate all the changes you promised in your revised version of the paper, especially the new comparison with the weighted sum version of self-consistency as a baseline in all models tested on all benchmarks included in the paper.

---

> > > > > ### Author Response · Authors · 2025-08-07
> > > > >
> > > > > Many thanks for engaging so thoughtfully with our rebuttal. We will integrate all the suggested changes in the final submission.

---

### Official Review · Reviewer_7zkf · 2025-07-01

**Clarity:** 2
**Significance:** 2
**Originality:** 2
**Rating:** 4
**Confidence:** 4

**Summary:**

The paper explores improving LLM performance for reasoning tasks by combining the results of N runs. Different methods of best-of-N strategies are discussed, with the suggestion that using self-certainty, a metric based directly on the token distributions in the LLM output, instead of just using the final tokens, is the best path. Voting is explored as well, combining the token distributions from several runs - similarly to directly using results from several runs. Some metric strategies are compared to the chosen one, in addition to baseline approaches based on external relevance models.

**Questions:**

1. How would I choose the parameter p?
2. I don't think the issue of resources was addressed adequately. If anything, assuming I understood the proposal and numbers correctly, I would suggest rewriting the paper as a proposal to gain most of the benefit of the best-of-N approaches by using the probability tokens of a single run (without voting).
3. Please add to table 1 description the specific metric used (accuracy?), and confidence intervals or at least a mention of statistical significance of results.
4. Is it really practical to analyze values such as N=64? I can't imagine running each task via an LLM X64 times in any applied scenario. If the KL-div metric only outperforms others when N>16, is it really more useful in practice?

**Ethical Concerns:**

["NO or VERY MINOR ethics concerns only"]

**Final Justification:**

The narrative in the paper seems somewhat broken and unclear, with a large emphasis on equations and less emphasis on explanations. However, the work has novelty and provides an interesting direction to a very important problem, so my final vote would be a borderline accept.

**Limitations:**

I would add the resource concerns to the limitation discussion.

**Paper Formatting Concerns:**

No major concerns.

**Quality:**

3

**Strengths And Weaknesses:**

The paper deals with an important problem and shows that best-of-N is a good path to improve performance, even if costly in compute/LLM resources. Several datasets are used in the analysis, as well as an adequate number of baselines and metric strategies.

The main weaknesses of the paper, in my view, are:
1. The narrative is very unclear, needlessly defining different metrics that only differ by a constant (self-certainty, KL-divergence and Cross Entropy). This means that the results are less readable, as sometimes they discuss KL-divergence and sometimes self-certainty. Moreover, the paper begins with the definitions of several metrics, some of which are considered as baselines and some as the method proposed by the paper. I would suggest a rewrite of the paper, focusing on first describing the specific proposed strategy - one metric, together with a better description of the voting method. Then the analysis and experimental sections can suggest the different baselines to compare against, as well as the voting method ablation.
2. Results are not clearly outperforming the baselines. No confidence intervals are presented. Using the probabilities of a single run does not seem to improve on the performance of best-of-N strategies. The gains from using the probabilities of multiple runs with voting, as opposed to just using best-of-N, seem minor. It is also unclear what would be a practical way of choosing the hyper-parameter p in the voting strategy, which seems to cause results to be sub-optimal. In general, the proposed strategy seems like a lot of effort for little-to-no gain.

---

> ### Author Rebuttal · Authors · 2025-07-31
>
> Thank you for your thoughtful feedback. We address each point below and will incorporate these clarifications in the revised manuscript.
>
> 1. **Clarity of Narrative (Weakness1 & Problem2,3)**
>
>    The overall flow of our paper is as follows: Our experiments are exploratory in nature rather than aimed at proving a specific theoretical result. We observed that most existing confidence measures rely solely on the probability of sampled tokens—such as perplexity—while discarding the distribution of unsampled tokens. This led us to ask an important question: **Does the discarded distribution encode useful information about the model’s confidence?**
>
>    To answer this, we introduce four candidate confidence measures that incorporate distributional information. Among these, KL-divergence performs best. We term this approach **self-certainty (SCe)** and adopt it as our primary metric.
>
>    To further enhance the performance of existing metrics, we propose **Borda voting** and evaluate both its effectiveness and the generalizability of self-certainty. We acknowledge that the switch between the term *KL* and *self-certainty* may reduce readability. To address this, we will revise the manuscript to consistently denote it as KL (SCe)* Additionally, we will clearly separate our main proposed method from baseline methods in the revised version for improved clarity.
>
>    Your suggestion to **separate probability-based methods and voting-based methods** to better highlight the strengths of our approach is greatly appreciated. We will reorganize the presentation accordingly and include the metric used (mean accuracy) in the tables for transparency.
>
> 2. **Significance of Results (Weakness2 & Problem3)**
>
>    We agree that emphasizing the statistical significance of results strengthens the paper. In the revised manuscript, we will include a **one-sided paired t-test** with the hypothesis that the proposed voting methods outperform self-consistency at fixed $N$ and $p$ across runs and datasets.
>
>    The p-values are shown below:
>
>    | Variant         | p (N = 8) | p (N = 64†) |
>    | --------------- | --------- | ----------- |
>    | Borda (p = 0.3) | 0.00      | 0.09        |
>    | Borda (p = 0.7) | 0.01      | 0.02        |
>    | Borda (p = 1.2) | 0.02      | 0.00        |
>    | Borda (p = 2.0) | \*\*      | 0.00        |
>
> † LiveBench-Math is excluded as the result is unavailable.
> \*\* Underperforms self-consistency.
>
> As shown, Borda voting with most values of $p$ **significantly outperforms self-consistency**.
>
> 3. **Choosing $p$ (Weakness2 & Question1)**
>
> We have discussed $p$-selection in the end of Section 6.2. Two practical strategies can be used when exhaustive search is undesirable:
>
> * **Rule of Thumb**: Use $p = 0.3$ for $N \leq 16$; $p = 1.2$ for larger $N$.
> * **Data‑Driven Tuning**: Tune $p$ on a held‑out validation set or a small subset of the test set.
>
> While these heuristics may not yield the global optimum, they still deliver performance that **significantly exceeds self-consistency**, as shown by the significance tests above.
>
> 4. **Practicality of Using $N > 16$ (Question 4)**
>
>    It is standard practice in the literature for best-of-$N$ methods to evaluate with $N > 16$. For example:
>
>    * Self-consistency \[1] uses $N = 40$.
>    * Scaling test-time compute considers $N = 512$ \[2].
>    * PRM evaluates up to $N = 1000$ \[3].
>
>    To contextualize $N = 64$: if a reasoning model requires 8× the tokens of a standard model, sampling 64 responses for a standard model is equivalent to \~8 samples for a reasoning model—a reasonable computational budget.
>
>    Furthermore, **strong performance at larger $N$ demonstrates the stability of self-certainty**, an important property for confidence metrics. This indicates robustness when handling more tokens with fewer errors, which is highly desirable.
>
> [1]Self-Consistency Improves Chain of Thought Reasoning in Language Models
>
> [2]Scaling LLM Test-Time Compute Optimally can be More Effective than Scaling Model Parameters
>
> [3]Let’s Verify Step by Step

---

> > ### Comment · Reviewer_7zkf · 2025-08-06
> > **Thank you for the response**
> >
> > I've reviewed the manuscript again and still have concerns about the narrative, which I still don't find understandable and don't see as a quick fix. Since my other concerns are mitigated, and I see value and novelty in the paper, I am updating my vote accordingly.

---

> > > ### Author Response · Authors · 2025-08-07
> > >
> > > Thank you for reevaluating our manuscript and acknowledging its value and novelty. We take your narrative concerns seriously and will revise the paper to improve clarity. In addition to the changes listed above, we will (i) add a roadmap paragraph at the end of the Introduction and (ii) place brief orientation statements at the start of each major section, so the paper’s structure and argument are immediately clear to the reader.

---

### Official Review · Reviewer_8gKi · 2025-07-01

**Clarity:** 2
**Significance:** 3
**Originality:** 3
**Rating:** 4
**Confidence:** 4

**Summary:**

This paper proposes a metric *self-certainty*, which measures the divergence of the uniform distribution and the output distribution over vocabulary. Then, it uses *self-certainty* with Borda Voting to conduct best-of-N to achieve effective test-time scaling.

**Questions:**

* Borda Voting may result in ties. How do you solve it?
* Can you provide the upperbound performance, label it correct as long as one of the N answers is correct?
* Which model precision do you use? I find that bf16 will cause a large bias in calculating self-certainty score.

**Ethical Concerns:**

["NO or VERY MINOR ethics concerns only"]

**Final Justification:**

All of my concerns are properly addressed. So I decide to raise my overall rating.

**Limitations:**

Yes

**Paper Formatting Concerns:**

No.

**Quality:**

2

**Strengths And Weaknesses:**

**Strength**:
* The proposed *self-certainty* proves to be a better confidence measure than perplexity, especially in terms of the unanswerable situations.
* It doesn't cost extra computation to get the *self-certainty* measure.

**Weakness**:
* *self-certainty* is not a normalized scalar. This limits the application of this metric. For example, it's hard to apply *self-certainty* to weighted average methods, as standard confidence scores can do[1]. I guess this is a reason for using Borda Voting, a not very suitable method. If the authors can figure out a way to normalize *self-certainty*, it will be a perfect solution.
* Borda Voting might not be a good voting method, because it discards the absolute values and only keeps the ordinal information. Besides, it shows instability when scaling, the hyperparameter (*p*) varies with *N*.
* There seems to be a conflict in experimental setting. While Line 219~220 explains Figure 4 with "*no-answer responses*", Line 170 states that "*Responses without extractable answers are masked*".
* Figure 2 is problematic. According to the definition of Self-Certainty, it should be irrelevant with the output token. Therefore, the first token of Sample I and II should have the same *self-certainty* score, but they are not. I doubt if these number are made up by human. Besides, Line 131 states "*sample I contains an initial error*", but I can't see why the first token "We" is an error.

[1] Efficient Test-Time Scaling via Self-Calibration.

---

> ### Author Rebuttal · Authors · 2025-07-31
>
> Thank you for your valuable feedback. Below, we address each of your concerns individually:
>
> 1. **Normalization of Self-Certainty (Weakness1)**
>
>    The method of normalizing self-certainty depends heavily on the specific use case and is typically straightforward. For example, if self-certainty is used in reinforcement learning, **z-score normalization** ($\frac{\text{sce} - \text{mean(sce)}}{\text{std(sce)}}$) can be an effective choice. Alternatively, if one intends to interpret self-certainty as a probability or scale it to the $[0, 1]$ range, **min-max normalization** ($\frac{\text{sce} - \min(\text{sce})}{\max(\text{sce}) - \min(\text{sce})}$) is applicable.
>
>    It is worth noting that even well-established metrics like probabilities often require transformation prior to application. For instance, in the self-calibration metric [1] you mentioned, a transformation of the form $\mathrm{SSC}(y)= \frac{\sum_{i : y_i = y} c_i}{\sum_{i=1}^{N} c_i}$ is employed. In this context, $c_i$ could seamlessly be replaced with self-certainty scores $sce_i$ for normalization purposes.
>
>    Furthermore, many widely used metrics in NLP, such as **perplexity** or **average log-probability**, are not normalized but are nonetheless informative and accepted by the community.
>
> 2. **Intuition Behind Borda Voting (Weakness1 & 2)**
>
>    Borda voting is a deliberate design choice, not a workaround for unnormalized scores. It balances the influence of self-certainty and self-consistency while remaining stable as $N$ changes. Table2 shows that simple sum voting performs between pure self-consistency and Borda voting. In small or highly imbalanced samples (e.g., $0.5+0.5>0.9$ or $0.5\times10>0.9\times5$), sum voting collapses to vanilla self-consistency, discarding confidence information. Borda voting mitigates this by ranking the scores and by letting us modulate the relative weight of confidence via the exponent $p$. Designing a scheme that keeps absolute scale, preserves stability across $N$, and offers the same control remains a promising avenue for future work.
>
> 3. **Clarification of Lines 219–221**
>
>    Thank you for pointing this out. The statement:
>
>    > *“Notably, perplexity assigns higher confidence to no-answer responses—often resulting from self-repetition or early stopping—while self-certainty reliably assigns these responses lower confidence scores.”*
>
>    was intended to reference Figure1, where we distinguish between correct, incorrect, and no-response cases, not Figure 4. We will revise the manuscript wording to eliminate this ambiguity.
>
> 4. **Clarification Regarding Figure 4**
>
>    We appreciate your close reading and catching the typo in Equation (2) and the formulas afterwards; it should be $p(\cdot|x, y_{\leq i})$ instead of $p(\cdot|x, y_{< i})$.Correspondingly, our token‑wise self‑certainty for the first token should measure the KL divergence **after** sampling token 1, which better reflects the model’s change in belief after deciding on one token. While this shift has minimal effect on the sentence‑wise self‑certainty (since it averages over many tokens), it yields more interpretable token-level values.
>
>    The values reported in Figure 4 are empirical, not made up. We examined multiple completions and selected representative examples based on their confidence trajectories. By “initial error” we meant the first reasoning step (step 1), not the first token. Our goal was to illustrate the model’s early preference between two reasoning patterns; we agree that aggregate metrics are more informative overall, and we will both correct the typo and clarify our intent when discussing Figure 4.
>
> 5. **Additional Questions**
>    * **Handling Ties:** In the rare event of a tie, we select the first sampled candidate. However, due to the use of the power function in weighting, ties are extremely uncommon in practice.
>
>    * **Upper Bound:** The upper bound of performance is provided in Appendix A.1, Figure 9, under the label *Oracle*, which significantly outperforms all mentioned metrics.
>
>    * **Model Precision:** The model uses **bfloat16** for weights during inference and **float32** for subsequent logit processing.
>
> We appreciate your insightful comments and believe the revised manuscript will reflect these improvements more clearly.
>
> [1] Efficient Test-Time Scaling via Self-Calibration.

---

> > ### Comment · Reviewer_8gKi · 2025-08-03
> >
> > Thank you for the detailed clarification.
> >
> > I still have problems on some of your responses.
> >
> > 1. **Normalization of Self-Certainty**
> >
> > Your explanation on normalizing a group of self-certainty values is clear. However, I am wondering if self-certainty is applicable to a single self-certainty value. Specifically, I want to know how you determine the $max(\text{sce})$ in min-max normalization? The self-certainty value grows exponentially with the probability of the top-probable token. Theoretically, $max(\text{sce})$ can be as big as infinite (in one-hot distribution), so I don't know how to choose the $max(\text{sce})$ empirically. Can you provide me with some suggestions?
> >
> > 2. **Intuition Behind Borda Voting**
> >
> > The results in Table 2 seem to convey a conclusion that a larger *N* best fits in a larger *p*. Does this conclusion hold? Can you provide any suggestions on the selection of *p*?
> >
> > 3. **Figure 2 is problematic**
> >
> > My initial review was not questioning any of your experiment results, but the example you used to draw Figure 2. I have no idea why you make clarifications on Figure 4.
> >
> > With your clarification on Equation (2), my first concern is addressed. The remaining one is *Line 131 states "sample I contains an initial error", but I can't see why the first token "We" is an error.*

---

> > > ### Author Response · Authors · 2025-08-04
> > >
> > > Thank you for your thoughtful feedback. We’ve addressed each point below:
> > >
> > > ### **1. Normalizing a Single Self-Certainty Value**
> > >
> > > Although self-certainty scores are unbounded in theory, in practice they tend to fall within a consistent range. For instance, in Figure 1 the Llama-3.1-8B-Instruct model produces self-certainty values between roughly 10 and 22.5, forming a bell-shaped distribution. To map any new score $x$ into $[0,1]$, one may apply percentile-based clipping followed by min-max normalization [1]:
> > >
> > > 1. **Estimate percentiles.**
> > >    Compute the 5th percentile $q_{0.05}$ and 95th percentile $q_{0.95}$ over representative sample of self-certainty scores.
> > > 2. **Clip outliers.**
> > >
> > >    $$
> > >      x' = \min\bigl(\max(x,\,q_{0.05}),\,q_{0.95}\bigr)
> > >    $$
> > > 3. **Normalize.**
> > >
> > >    $$
> > >      x_{\text{norm}} = \frac{x' - q_{0.05}}{q_{0.95} - q_{0.05}}
> > >    $$
> > >
> > > This “percentile clipping” approach ensures that extreme values do not distort the normalized scale while preserving the central tendency of the model’s confidence distribution.
> > >
> > > ### **2. Borda Voting and the Choice of $p$**
> > >
> > > The observed relationship—that larger $N$ aligns with larger optimal $p$—is consistent across our dataset. As shown in Table 1 and Figure 7, the optimal $p$ increases as $N$ grows.
> > > Regarding practical selection of $p$:
> > >
> > > * **Rule of Thumb:** Use $p \approx 0.3$ for $N \leq 16$ and $p \approx 1.2$ for larger $N$.
> > > * **Data‑Driven Tuning:** When feasible, tune $p$ on a validation set or a small fraction of the test set. In our experiments, convergence of optimal $p$ typically occurs with about 1,000 samples.
> > >   When exhaustive search is impractical, these strategies provide an effective balance between performance and efficiency.
> > >
> > > ### **3. Clarification on Figure References**
> > >
> > > We apologize for the earlier mix-up. In point 4 of our previous response, all references should have pointed to Figure 2, not Figure 4. Furthermore, by “initial error” we meant the error at the **first reasoning step**, not the first token. Our inclusion of first-token metrics was intended only to illustrate the model’s early pattern preference; the stepwise metrics (as shown in Figure 2) are indeed the most relevant for evaluating reasoning quality. We will update the manuscript to make these distinctions clear.
> > >
> > > [1]Tukey, J. W. (1977). Exploratory Data Analysis. Addison‐Wesley.

---

> > > > ### Comment · Reviewer_8gKi · 2025-08-05
> > > >
> > > > I thank the authors for the clarifications.
> > > >
> > > > I am still confused by the third point. While you claim that the "initial error" refers to the **first reasoning step**, Line 134 states " distributional confidence detects correct reasoning **from the first token**". Can you explain this clearly?

---

> > > > > ### Author Response · Authors · 2025-08-06
> > > > >
> > > > > Thank you for pointing that out. You’re right that the sentence in Line 134 is problematic. Our intent was to convey that, in Sample I, the model jumps straight to factoring without first checking whether the expression is factorable, which is a skip-step behavior. By contrast, in Sample II, the model recognizes the quadratic form and selects the quadratic formula as the safer approach. We had hypothesized that prompts beginning with “We” might induce skip-step behavior because they often launch directly into solving steps, whereas prompts beginning with “This” encourage more careful examination. We agree that this claim, as stated, is neither intuitive nor rigorous. We will revise our analysis to refer to the “first few tokens” rather than the “first token” and update the manuscript accordingly. Thank you again for catching this.

---

> ### Comment · Reviewer_8gKi · 2025-08-06
>
> Thank you for this explanation.
>
> Now I fully understand what this example wants to convey. In your revised version, please either explain it clearly or use another straightforward case.
>
> I am generally satisfied with this paper so far, and I have one last concern about the generalizing ability of self-certainty. In Table 1, self-certainty improves over self-consistency by using a better confidence measurement. On the other hand, the Soft Self-Consistency (SSC) [1] method we discussed earlier improves over self-consistency using a more delicate answer aggregation method. So, it comes to me that whether combining self-certainty and the SSC method yields even better improvement.
>
> [1] Efficient Test-Time Scaling via Self-Calibration.

---

> > ### Author Response · Authors · 2025-08-07
> >
> > Thank you for raising this excellent point. Applying soft self-consistency with self-certainty to response selection is equivalent to sum voting after canceling the common denominator. As shown in Table 2, this approach is less flexible than Borda voting. However, combining soft self-consistency and self-certainty for other applications—such as improving LLM calibration, as discussed in the self-calibration paper you referenced—is a promising direction for future research, especially in light of the evidence you highlighted. We will incorporate this into the future work section of the revised manuscript and clarify the example we previously discussed.

---

> > > ### Comment · Reviewer_8gKi · 2025-08-07
> > >
> > > Thank you so much for the clarification! I have no more questions and I will raise the overall rating.

---

### Official Review · Reviewer_3kq2 · 2025-07-03

**Clarity:** 4
**Significance:** 3
**Originality:** 2
**Rating:** 4
**Confidence:** 4

**Summary:**

This paper proposes a lightweight and scalable approach for Best-of-N selection in Large Language Models (LLMs) based on a novel metric called self-certainty. Unlike reward model-based or self-consistency methods, self-certainty quantifies the model’s internal confidence by computing the divergence of the output token distribution from a uniform distribution. The authors show that this distributional confidence measure can distinguish correct from incorrect responses more effectively than traditional log-probability or perplexity-based metrics. To improve performance on tasks with explicit answers, they introduce a self-certainty-based Borda voting scheme that weights responses by confidence rank. The method is evaluated across mathematical reasoning, code reasoning, and code generation tasks, demonstrating consistent improvements over self-consistency, Universal Self-Consistency (USC), and greedy decoding. The results highlight the method's scalability, generalizability to open-ended tasks, and effectiveness in selecting higher-quality responses without additional training or inference cost.

**Questions:**

1. On many tasks, self-certainty without Borda underperforms self-consistency, suggesting it may not be useful without Borda voting. The authors need to also conduct evaluation on Borda voting with other confidence measures, such as KL, Entropy, DP, SSC, to show the effectiveness of self-certainty.

**Ethical Concerns:**

["NO or VERY MINOR ethics concerns only"]

**Final Justification:**

Most of my concerns are solved in rebuttal. The only remaining concern is what is the basis for choosing $p=0.3$ (for example, why not 0.4 or 0.5?) for $N \leq 16$. I hope the author can make the changes as promised in further version.

**Limitations:**

Yes

**Quality:**

3

**Strengths And Weaknesses:**

**Strengths**:
1. The paper introduced a new metric called self-certainty and a self-certainty-based Borda voting scheme.
2. The experimental results show that the proposed method consistently outperforms various baselines, such as self-consistency, universal self-consistency across diverse benchmarks.
3. The paper is well-structured and clearly written.

**Weaknesses**:
1. Soft self-consistency (SSC) [1] use aggregation of probability of each token to conduct the Best-of-N selection also challenges limitations of traditional self-consistency methods on open-ended generation tasks. This paper should discuss and compare with this method.
2. The proposed method requires access to token probability, but the most performant LLMs are typically black-box models often with limited or no access to logits. It can not generalize to these black-box models. Therefore, some work has explored prompting language models to express uncertainty or confidence score in human language [2,3]. I think this paper should discuss and compare with this kind of method.
3. On many tasks, self-certainty without Borda underperforms self-consistency, suggesting it may not be useful without Borda voting. The authors need to also conduct evaluation on Borda voting with other confidence measures, such as KL, Entropy, DP, SSC, to show the effectiveness of self-certainty.
4. The Borda voting method requires tuning a hyperparameter $p$ to balance confidence and frequency, and optimal values vary across datasets and sample sizes, which may hinder out-of-the-box applicability.

[1] Soft Self-Consistency Improves Language Models Agents (Wang et al., ACL 2024)

[2] Just Ask for Calibration: Strategies for Eliciting Calibrated Confidence Scores from Language Models Fine-Tuned with Human Feedback (Tian et al., EMNLP 2023)

[3] Can LLMs Express Their Uncertainty? An Empirical Evaluation of Confidence Elicitation in LLMs (Xiong et al., ICLR 2024)

---

> ### Author Rebuttal · Authors · 2025-07-31
>
> Thank you for your thoughtful review! We’re grateful for the opportunity to clarify and strengthen our manuscript.
>
> 1. **Comparison with Soft Self‑Consistency (Weakness1)**
>    Thank you for pointing out the Soft Self‑Consistency paper—it’s highly relevant. We have now directly compared our self‑certainty measure against soft self‑consistency (using mean‑aggregation over the entire response as the action space) on the MATH‑Level 4 dataset with $N=16$. Results:
>
>    | Method       | First Answer | Self‑Certainty | Soft Self‑Consistency |
>    | ------------ | -----------: | -------------: | --------------------: |
>    |   Accuracy  |       57.21 |         64.37 |                58.53 |
>
>    These results indicate that self-certainty significantly outperforms soft self-consistency under equivalent settings. While the original soft self-consistency also introduces the idea of averaging over the answer space alone (instead of the full response), this innovation is equally applicable to self-certainty.
>
>    Moreover, in scenarios with sparse answer spaces—such as those discussed in the SSC paper—self-certainty can be especially advantageous over self-consistency. This is because each response is more likely to produce a unique answer group, reducing the effectiveness of self-consistency. We will cite and discuss the Soft Self-Consistency paper in the next version of the manuscript.
>
> 2. **Usefulness of Self‑Certainty (Weakness2 & 3)**
> Although self‑certainty alone underperforms full self‑consistency methods, it offers several unique advantages:
>
>    * **Fine‑Grained Confidence**
>    Self‑certainty provides token‑level confidence scores, enabling potential of “local” or process‑based scoring.
>
>    * **Generalizability**
>    Unlike self-consistency, which relies on comparable final answers, self-certainty is applicable to a wider range of tasks, including open-ended or generative tasks such as free-form dialogue. It is particularly useful in cases where the answer space is vast or difficult to isolate from the overall response. In addition, self-certainty can be integrated with GRPO to support reinforcement learning without external feedback.
>
>    * **Model Calibration**
>    Self-certainty is effective at capturing model calibration. As shown in Figure 1, it better separates correct from incorrect and null responses compared to perplexity. In Appendix A2, Figure 10, we show that self-certainty correlates well with question difficulty, indicating its potential as a proxy for model performance or task complexity in the absence of ground-truth answers.
>
>    * **Standalone Metric**
>       In contrast to composite methods that require multiple scoring mechanisms, self-certainty is a standalone metric. It can replace or complement traditional token probability–based confidence signals. For example, self-certainty can substitute the token-probability weights in soft self-consistency, potentially improving its performance.
>
>    * **Potential for Black-Box Models**
>    Although current APIs may not expose self-certainty, the metric does not require revealing the full predictive distribution, making it compatible with plausible future API designs (e.g., exposing token-level confidence summaries). Moreover, for black-box systems, one could approximate self-certainty using a white-box surrogate’s logits, which can be a promising future work.
>
> 3. **Voting Methods with Different Confidence Measures (Weakness3 & Question)**
>    We agree that a head-to-head comparison is helpful. On MATH-Level-5 with Llama-3.1-8B-Instruct (N=16), we evaluated voting using several confidence measures and Borda exponents p.
>
>    Method           | Perplexity | (D) Entropy | (D) Perplexity | (D) Gini | Self‑certainty |
>    | ---------------- | ---------: | ----------: | -------------: | -------: | -------------: |
>    | Self‑consistency (p=0) |      37.99 |       37.99 |          37.99 |    37.99 |          37.99 |
>    | Borda (p = 1)  |      38.60 |       39.12 |          38.14 |    39.35 |          38.90 |
>    | Borda (p = 2)  |      38.60 |       38.40 |          38.14 |    38.75 |          39.43 |
>
>    When $N$ and the exponent $p$ is large, self‑certainty–based Borda voting outperforms other metrics, which demonstrates its robustness. We will include this table and analysis in the revision.
>
> 4. **Choosing the Optimal Exponent $p$ in Voting (Weakness4)**
>    We appreciate the concern about selecting $p$. As discussed at the end of section6.2, the optimal
>    $p$ increases with the number of samples N. Except exhuastive search, below are two possible way of determine $p$ to use
>
>    * **Rule of Thumb**: use $p=0.3$ for $N\le16$; $p=1.2$ for larger $N$.
>    * **Data‑Driven Tuning**: one can also tune $p$ on a held‑out validation set or a small fraction of the test set.
>
>    We will expand this discussion and provide more guidance in the revised manuscript.
>
> Thank you again for your valuable feedback. We look forward to incorporating these revisions and believe they will significantly strengthen our work.

---

> > ### Comment · Reviewer_3kq2 · 2025-08-04
> >
> > Thank you for your detailed response. In your second table, when $p = 1$, the Entropy-based and Gini-based Borda voting methods outperform the Self-certainty-based variant. However, in your fourth point, you suggest using $p = 0.3$ for $N \leq 16$ and $p = 1.2$ for larger $N$ based on a rule of thumb. These two statements appear to be somewhat inconsistent. Could you provide a comparison across a more fine-grained range of $p$ values to clarify how performance varies with $p$?

---

> > > ### Author Response · Authors · 2025-08-04
> > >
> > > Thank you for the thoughtful and constructive feedback. Here is a more fine-grained version of the above table:
> > > | Method                 | Perplexity | (D) Entropy | (D) Perplexity | (D) Gini | Self‑certainty |
> > > | ---------------------- | ---------: | ----------: | -------------: | -------: | -------------: |
> > > | Self‑consistency (p=0) |      37.99 |       37.99 |          37.99 |    37.99 |          37.99 |
> > > | Borda (p = 0.5)        |      38.52 |       37.69 |          38.60 |    39.12 |          38.67 |
> > > | Borda (p = 1.0)        |      38.60 |       39.12 |          38.14 |    39.35 |          38.90 |
> > > | Borda (p = 1.5)        |      38.82 |       38.97 |          38.14 |    39.05 |          38.90 |
> > > | Borda (p = 2.0)        |      38.60 |       38.40 |          38.14 |    38.75 |          39.43 |
> > > | Borda (p = 2.5)        |      39.05 |       34.67 |          37.69 |    38.52 |          39.27 |
> > > | Borda (p = ∞)          |      27.34 |       23.64 |          22.43 |    23.87 |          31.42 |
> > >
> > > As shown, self‑certainty achieves its best performance at a relatively larger $p$ compared to other metrics, indicating that its ranking ability is stronger. As noted earlier, the heuristic “choose $p = 1.2$ for $N > 16$” is only a rough rule of thumb for convenience when one does not wish to tune hyperparameter. A better practice is to tune $p$ on a validation set or a small fraction of the test set. While this introduces some extra tuning cost, it is usually negligible compared to the overall application cost. We will clearly state this limitation and recommendation in the revised version of our paper.

---

> > > > ### Comment · Reviewer_3kq2 · 2025-08-04
> > > >
> > > > Thank you for your prompt response. The fine-grained results table shows that when $p$ is large, the Self-certainty–based Borda voting outperforms the other metrics. However, this does not fully justify the choice of $p = 1.2$ for $N > 16$ and $p = 0.3$ for $N \leq 16$ when hyperparameters are not tuned. Even if this choice is based on a rule of thumb, I would expect it to be supported by experimental results, but this connection is not evident from the table.

---

> > > > > ### Author Response · Authors · 2025-08-05
> > > > >
> > > > > Thank you for pointing this out. We refer to this heuristic for selecting $p$ as a “rule of thumb,” since it may not be optimal for every dataset or value of $N$. To validate our choice, we further evaluated Borda voting with these confidence measures on MATH-Level4 using $N=64$. The results are summarized below:
> > > > >
> > > > > | Method              | Perplexity | (D) Entropy | (D) Perplexity | (D) Gini | Self-certainty |
> > > > > | ------------------- | ---------: | ----------: | -------------: | -------: | -------------: |
> > > > > |   Borda (p = 1.2)   |      61.45 |       61.70 |          60.71 |    61.78 |          62.19 |
> > > > >
> > > > > Under these conditions, Borda voting with self-certainty ($p = 1.2$) outperforms the other metrics, supporting our selection of $p = 1.2$ as a reasonable default. In the revised manuscript, we will:
> > > > >
> > > > > 1. **Expand our comparison** to include confidence measures alongside Borda voting over more dataset for completeness.
> > > > > 2. **Clarify** that the “rule of thumb” should be applied with caution, as its effectiveness may vary across datasets and $N$.
> > > > > 3. **Highlight** that tuning $p$ via a simple grid search over a representative subset is inexpensive—taking only a few hours on a single GPU (Llama 3.1-8B-Instruct)—and thus practical even without optimized implementation.
> > > > >
> > > > > We will also refine our phrasing around the recommendation for choosing $p$ to eliminate any remaining confusion.

---

> > > > > > ### Comment · Reviewer_3kq2 · 2025-08-05
> > > > > >
> > > > > > Most of my concerns are solved in rebuttal. I hope the author can make the changes as promised in further version. I have increased my score accordingly.

---

> > > > > > > ### Author Response · Authors · 2025-08-07
> > > > > > >
> > > > > > > Thank you for your thoughtful review and for engaging with our responses. We appreciate your suggestions and will incorporate the discussed changes in the camera-ready version.

---

### Official Review · Reviewer_f3GF · 2025-07-05

**Clarity:** 2
**Significance:** 3
**Originality:** 2
**Rating:** 4
**Confidence:** 3

**Summary:**

The paper discusses existing techniques in best-of-N selection in LLMs, and introduces *self-certainty* — namely, the empirical KL divergence from the uniform distribution over the tokens dictionary to the corresponding predictive conditional probability of the model averaged over the generated output — as a scalable metric to improve predictive performance in the absence of external rewards. Authors empirically showcase that self-certainty scales better with more samples, is free from output length biases, and suitable for open-ended generation, demonstrating better performance from competitive methods via the crucial addition of a Borda voting scheme. Experiments are carried out on mathematical and code reasoning datasets through the use of open-source 8B models.

**Questions:**

* The voting scheme seems to be boosting significantly the self-condifence based performance, however this add-on is not considered for the other baselines: Have authors experimented with adding the voting scheme to these as well?

* Why do baselines fail to improve with sample size N≥16 as showed in Fig.4?

* Another missing dimension is to examine how the performance of the new metric correlates
with the calibration of the LLM. Are distributional divergence based metrics better candidates for best-of-N selection in miscalibrated models, or do they get more easily fooled by overconfident output as hinted in Section 4?

* The message of Fig. 2 is unclear to me: how can the displayed numbers be compared across the two methods? Is there a way to normalize the considered metrics so that they are legitimately comparable?

* While a convincing case is made towards the advantages of the introduced metric in analysing long output, it's unclear what are the limitations of self-certainty when facing short output. Can multiple-choice QA benchmarks be included in the experimentation section? Can voting successfully address distortion due to high confidence scores for small number of outputted tokens?

* The presentation would also be benefited by including an analysis of the runtime complexity overhead and inference-time memory requirements for the analysed best-of-N selection methods.

**Ethical Concerns:**

["NO or VERY MINOR ethics concerns only"]

**Final Justification:**

After authors response, I'm still weakly leaning towards paper's acceptance. I think extra experiments on voting applied to baselines and multiple choice benchmarks can be included in the revised version, offering a fuller picture of the proposed metric.

**Limitations:**

Yes

**Quality:**

3

**Strengths And Weaknesses:**

**Strengths**

The major strength of the paper is the introduction of the self-certainty metric: Replacing log-likelihood of generated tokens with a metric accounting for distributional confidence over the supported dictionary, the new distributional metric is able to capture more holistic information and deal with some inherent statistical weaknesses of maximum-likelihood based metrics. Experiments are convincing in displaying the premise of better behavior of the new metric over lengthy and reasoning-intensive output generations.

**Weaknesses**

On the other hand, there is insufficient theoretical justification regarding why the new metric works better and an underwhelming interpretation of observed failure cases. It’s unclear why Kullback-Leibler (KL) was preferred from other statistical divergences, why the arguments of KL (an asymmetric divergence) where selected in this order, and
why divergence from the uniform is included, as it only adds a constant term to the metric. Moreover, according
to the demonstrated results in Table 1, the Borda voting contributes crucially to the achieved performance, while self-certainty without voting is mostly subpar to other competitive method when this component is not included.

---

> ### Author Rebuttal · Authors · 2025-07-31
>
> We sincerely appreciate your thoughtful feedback. Below, we address your specific concerns and suggestions:
> 1. **Justification of Choosing Reverse KL with Uniform Distribution as Main Metrics (Weakness)**
>
>     We chose reverse KL divergence to a uniform distribution as our primary confidence metric based on extensive exploratory experiments.  Our motivation was to ask: *“Does the probability mass beyond the sampled token help us assess model confidence for best‑of‑N selection?”* We compared several candidates (e.g., forward KL, entropy, perplexity) and found that reverse KL is the most robust, particularly as N increases: it does not suffer from the same scaling “pitfalls” as other metrics.
>
>     Regarding the uniform reference, we also tried using the empirical distribution over the N samples (Appendix A.4) and observed no significant difference. We retain the constant term in the uniform‐KL for easier interpretation; users may omit it if desired.
>
>    To further elucidate the behavior of “self‑certainty,” we will include two formal properties (and their proofs) in the next revision:
>
>    1. **SFT increases self‑certainty** on the training completions.
>    2. **Self-certainty is less prone to repetition trap**. When the repetition‑branch distribution features a flatter tail, the self‑certainty–preferred collapse set is a subset of the distributional entropy or perplexity sets.
>
>
> 2. **Incorporating a Voting Scheme over Multiple Metrics (Question1)**
>
>    We agree that combining confidence metrics can be beneficial.  We have implemented a simple Borda‐count voting scheme on MATH‑level 5 (N = 64) and report preliminary results below:
>
>    | Method           | Perplexity | (D) Entropy | (D) Perplexity | (D) Gini | Self‑certainty |
>    | ---------------- | ---------: | ----------: | -------------: | -------: | -------------: |
>    | Self‑consistency (p=0) |      37.99 |       37.99 |          37.99 |    37.99 |          37.99 |
>    | Borda (p = 1)  |      38.60 |       39.12 |          38.14 |    39.35 |          38.90 |
>    | Borda (p = 2)  |      38.60 |       38.40 |          38.14 |    38.75 |          39.43 |
>
>    These results show that, for large N, self‑certainty dominates when the Borda exponent p is tuned appropriately—demonstrating its robustness relative to other metrics. We will include this table in the revised manuscript.
>
> 3. **Why the “FirstAns” Baseline Stops Improving for N ≥ 16 (Question2)**
>
>    The “FirstAns” baseline simply selects the first valid answer encountered among the N samples.  In our experiments, all questions already yield a valid answer within the first 8 samples, so increasing N beyond 8 offers no benefit.  Other metrics degrade for large N because they can be misled by over‑confident incorrect responses.
>
> 4. **Self‑certainty as a Measure of Model Calibration (Question3)**
>
>    Indeed, self‑certainty serves as an effective proxy for calibration.  In Figure 1, self‑certainty cleanly separates correct responses from no-response and incorrect ones, whereas perplexity exhibits weaker separation.  Furthermore, Appendix A.2 (Figure 10) shows that average self‑certainty declines significantly as problem difficulty increases, matching the drop in accuracy.  We will make these calibration connections explicit in the revised draft.
>
> 5. **Clarification on Figure 2 Normalization (Question4)**
>
>    In Figure 2, we compare each metric against itself across two sample sets—not the two metrics against each other—so no additional normalization is needed.  Both self‑certainty and perplexity are already length‑normalized by definition, allowing direct comparison across samples of varying lengths.
>
> 6. **Runtime Complexity and Memory Overhead (Question6)**
>
>    Computing self‑certainty incurs approximately no extra cost: the logits required are already produced during sampling, so best‑of‑N strategies mentioned in our manuscript (regardless of confidence metric) share the same time and memory complexity, which is dominated by the sampling process itself.  While one could leverage self‑certainty for dynamic sampling given the roboustness demonstrated in our article, that extension is beyond this paper’s scope.
>
> 7. **Self-certainty on Short Questions (Question5)**
>    Since self-certainty is a logit-based method, we infer that for short questions there are too few logits to average over, and the task relies more on memorization than on reasoning, self‑certainty’s effectiveness can decline. Here is an experiment on a Commonsense QA using Llama 3.1‑8B‑Instruct with N = 8, we obtain:
>
>    | Method                 | Accuracy (%) |
>    | :--------------------- | -----------: |
>    | FirstAns.              |        51.02 |
>    | Self‑consistency       |        59.05 |
>    | Self‑certainty         |        53.64 |
>    | Borda voting (p = 0.7) |        60.11 |
>
>    Here, Borda voting still able to outperform self-consistency under the preliminary setting. We will include the result of the multiple choice QA dataset for completeness in the future revision.
>
>
> We trust these clarifications and the additional results will address your concerns. Thank you again for the valuable feedback!

---

### Decision · Program_Chairs · 2025-09-17

**Decision:**

Accept (poster)

**Comment:**

This paper introduces "self-certainty," a lightweight metric for Best-of-N selection in LLMs. The method leverages the model's internal token probability distribution to estimate response quality, offering a computationally efficient alternative to resource-intensive reward models.

The core of the proposed method is to quantify the divergence of the LLM's output distribution from a uniform distribution (using KL-divergence) as a measure of confidence. This "self-certainty" score is then used to select the best response from N generated samples. The authors also propose a Borda voting that combines self-certainty rankings with response frequency to improve performance on tasks with definite answers. The work is relevant to NeurIPS, addressing the challenge of improving the reasoning and reliability of LLMs, a central topic in the field. Specifically, it contributes to the area of efficient inference-time optimization for language models.

The primary innovation lies in using the entire token distribution for confidence scoring, rather than just the probability of the generated tokens (like perplexity). This provides a more robust signal of the model's confidence. The application of a Borda voting to integrate this confidence score with self-consistency is also a fine contribution.

Tihs paper received consistent ratings from all five reviewers, with each assigning a score of 4 (Borderline accept).
Also, confidence levels were generally high. There were no significant conflicting viewpoints among the reviewers. All converged on a similar assessment of the paper's strengths and weaknesses.

Reviewers acknowledged the novelty and potential of the self-certainty metric. They highlighted its efficiency (minimal/near-zero overhead), scalability with the number of samples (N), and its applicability to open-ended tasks where traditional self-consistency fails. The paper is also well-structured and clearly written.

A primary concern shared by reviewers was the observation that the self-certainty metric alone often underperforms the self-consistency baseline. The strong performance of the proposed method appears to be heavily reliant on the Borda voting. Reviewers pointed out that this voting mechanism was not applied to other baseline confidence measures, making a direct comparison of the self-certainty metric's effectiveness difficult. The need to tune the hyperparameter 'p' for the Borda voting was also noted as a potential limitation. Also, multiple reviewers note that SCe without Borda often trails self-consistency - the gains primarily come from pairing SCe with Borda.

While the paper provides a strong intuition for why distributional confidence should be a better metric, reviewers noted a lack of deep theoretical justification for the specific choice of KL-divergence from a uniform distribution. The authors' rebuttal promised to include formal properties and proofs in the final version.

Extensive experiments are presented across multiple reasoning benchmarks (mathematical, code reasoning, code generation). The results demonstrate that the self-certainty-based Borda voting consistently outperforms baselines. However, the ablation of the Borda voting component reveals that self-certainty alone is not always superior, which is a key point of discussion.

Overall, the paper introduces a novel and intuitive idea to address the important problem of Best-of-N selection in LLMs. The proposed method is simple, computationally efficient, and shows consistent empirical gains over strong baselines when combined with a voting mechanism. The work is well-executed and the authors' rebuttal effectively addressed the majority of the reviewers' concerns, demonstrating a clear path to an improved final version. The authors must incorporate the additional experiments, clarifications, and theoretical justifications promised in their rebuttal.